# Loss-of-function, gain-of-function and dominant-negative mutations have profoundly different effects on protein structure

Lukas Gerasimavicius [1], Benjamin J. Livesey [1] & Joseph A. Marsh [1]✉

Most known pathogenic mutations occur in protein-coding regions of DNA and change the way proteins are made. Taking protein structure into account has therefore provided great insight into the molecular mechanisms underlying human genetic disease. While there has been much focus on how mutations can disrupt protein structure and thus cause a loss of function (LOF), alternative mechanisms, specifically dominant-negative (DN) and gain-of-function (GOF) effects, are less understood. Here, we investigate the protein-level effects of pathogenic missense mutations associated with different molecular mechanisms. We observe striking differences between recessive *vs* dominant, and LOF *vs* non-LOF mutations, with dominant, non-LOF disease mutations having much milder effects on protein structure, and DN mutations being highly enriched at protein interfaces. We also find that nearly all computational variant effect predictors, even those based solely on sequence conservation, underperform on non-LOF mutations. However, we do show that non-LOF mutations could potentially be identified by their tendency to cluster in three-dimensional space. Overall, our work suggests that many pathogenic mutations that act via DN and GOF mechanisms are likely being missed by current variant prioritisation strategies, but that there is considerable scope to improve computational predictions through consideration of molecular disease mechanisms.

[1] MRC Human Genetics Unit, Institute of Genetics & Cancer, University of Edinburgh, Edinburgh, UK. ✉email: joseph.marsh@ed.ac.uk

Missense single nucleotide variants, which result in the substitution of a single amino acid residue at the protein level, are responsible for a large fraction of all currently known human genetic disorders[1,2]. Such disease-causing variants generally display a lower frequency of being observed in the population due to selective pressures. As a result of employing allele frequency filtering as an integral part of clinical genetics pipelines, a large number of rare variants that lack evidence of pathogenicity have been assigned as variants of uncertain significance. However, pathogenic missense variants represent only a small fraction of rare variants that have been observed in the human population, as all of us in fact harbour a significantly higher number of benign, but rare, variants than has been considered before[3–5]. Thus, separating causal missense changes that are truly clinically relevant from rare benign variation remains a major challenge for diagnosis, and ultimately treatment, of human genetic disease.

A large number of computational methods have been developed for the prediction of variant effects, and are widely used in clinical sequencing pipelines for the prioritisation of potentially damaging variants. Such variant effect predictors (VEPs) are diverse in their methodologies and implementation, leveraging protein sequences and/or structures, as well as various contextual annotations at the gene, protein or residue level[6,7]. While their current utility in a clinical setting is still limited by their insufficient accuracy[8], the improvement and application of VEPs is expected to remain a major avenue to variant prioritisation[9].

VEPs can provide useful information on the likelihood of mutations being pathogenic, but most tell us nothing about the molecular mechanisms underlying disease. For this, consideration of the protein structural context of mutations can be very informative. In particular, protein stability predictors, which directly evaluate the change in Gibbs free energy of folding ($\Delta\Delta G$) upon mutation, represent an alternate computational strategy for understanding the effects of missense mutations. Most stability predictors directly utilise protein structures to model the change in stability between the wild-type and variant proteins through a scoring function of pairwise atomic or coarse-grained interactions[10]. While these methods were not specifically designed for the identification of pathogenic variants, they are routinely used when evaluating candidate mutations[11–14] in order to identify those that are likely to be damaging to protein structure and thus cause a loss of function (LOF)[15–17]. Alternatively, increased protein stabilisation can also be associated with disease, and it has been shown that using the absolute $\Delta\Delta G$ values results in higher accuracy when identifying disease mutations, although this may also be due to predictor inability to correctly distinguish the direction of the effect[18,19]. Interestingly, a recent study found that stability predictors performed much better in the identification of pathogenic missense mutations in genes associated with haploinsufficiency[20], supporting the utility of stability predictors for identifying LOF mutations.

Although many pathogenic missense mutations cause a simple LOF, a large number are known to operate via alternate molecular mechanisms. For example, with the dominant-negative (DN) effect, the expression of a mutant protein interferes with the activity of a wild-type protein[21]. This is most commonly observed for proteins that form homomeric complexes, in which the mutant subunits can effectively "poison" the assembly[22]. Such mutations should not be highly destabilising, as the DN effect is reliant on the mutant protein being stable enough to co-assemble into a complex with the wild type. In fact, it has previously been observed, for a limited subset of transmembrane channel proteins, that DN missense mutations tend to have low predicted $\Delta\Delta G$ values[14]. Similarly, we can hypothesise that gain-of-function (GOF) mutations, which can occur through various mechanisms, such as constitutive activation, shift of substrate or binding target specificity, or protein aggregation[23], should also tend to be mild at a protein structural level.

In this study, we have investigated the effects of pathogenic missense mutations associated with different molecular disease mechanisms on protein structure. We find clear differences between LOF vs non-LOF mutations in terms of their location within structures, their predicted effects on protein stability, and their clustering in three-dimensional space. Most importantly, we find that nearly all the VEPs we tested perform worse on DN and GOF mutations, which shows that there are systematic limitations in the ability of current computational predictors to identify disease mutations associated with non-LOF mechanisms.

## Results

**Consideration of full protein complex structures improves the identification of disease mutations**. As the basis for this study, we first compiled a dataset of human missense mutations mapped to three-dimensional protein structures from the Protein Data Bank[24] of 1261 Mendelian disease genes (schematic outline of our data collection and annotation pipeline is outlined in Supplementary Fig. 1; complete dataset is accessible through the link in section 'Data Availability'). Due to the nature of the PDB, this dataset is focused on structured proteins, and will contain very few intrinsically disordered protein regions. The compiled set included 13,050 annotated pathogenic and likely pathogenic missense mutations from ClinVar[3], and 211,266 missense variants observed across >140,000 people from gnomAD v2.1[4]. We recognise that the gnomAD dataset will contain some damaging variants, e.g., those that are associated with late-onset disease, population-specific penetrance or are pathogenic under homozygous conditions. The vast majority of the gnomAD variants are classified as rare according to clinical genetics standards (<0.1% allele frequency), and performing allele frequency-based filtering of this dataset would drastically diminish the available data and statistical power of our analyses, despite most rare variants observed in the general population being benign or of sub-clinical significance[5]. Thus, we have chosen to include all the gnomAD variants (excluding those annotated as pathogenic in ClinVar) as our 'putatively benign' dataset.

Next, we modelled the effects of all missense changes using the structure-based protein stability predictor FoldX v5[25]. We used FoldX for three reasons. First, we previously showed it to have the best performance, out of 13 different stability predictors, in distinguishing between ClinVar pathogenic and gnomAD variants[18]. Second, most other stability predictors consider only the structures of individual polypeptide chains, and are incapable of evaluating stability changes on protein complexes, which is a key focus of our study. Finally, most stability predictors are only accessible as webservers, making it untenable to apply them to our very large dataset.

Based on our previous analysis, here we have primarily used absolute values, $|\Delta\Delta G|$, which show a better ability to discriminate between pathogenic and putatively benign variants. This could reflect two things. First, some pathogenic mutations might increase protein stability, and so the magnitude of the stability perturbation is most useful for identifying these stabilising disease mutations. Second, FoldX, as well as other similar stability predictors, may be better at predicting the magnitude than the sign of the stability perturbation. This was supported by the fact that different stability predictors are often discordant in whether or not a mutation is stabilising or destabilising[18]. Importantly, for most analyses, we found it makes very little difference whether we use raw or absolute $\Delta\Delta G$, as the large majority are destabilising. We will further address the issue of stabilising vs destabilising mutations later in the manuscript.

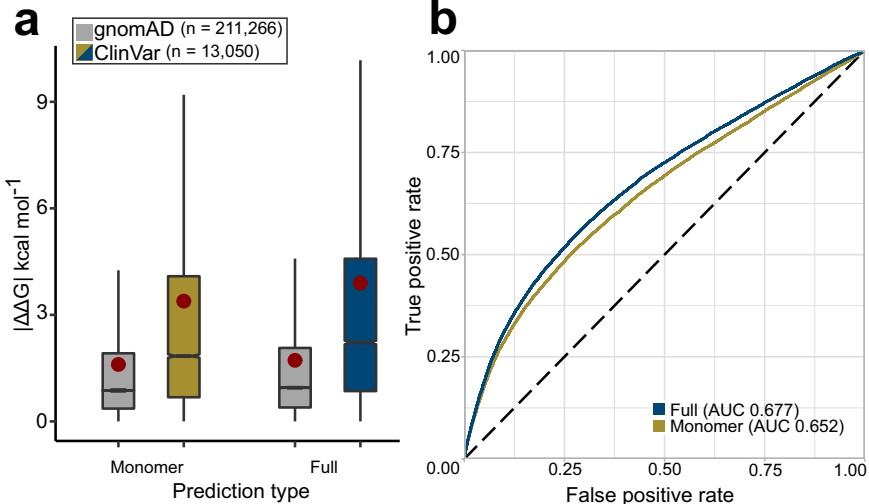

**Fig. 1 Using full protein complex structures improves discrimination between pathogenic and putatively benign missense variants. a** Comparison of predicted |ΔΔG| values for putatively benign gnomAD and pathogenic ClinVar mutations when calculated using monomer structures only, or by also employing full biological assemblies and therefore also considering intermolecular interactions. Boxes denote data within 25th and 75th percentiles, and contain median (middle line) and mean (red dot) value notations. Whiskers extend from the box to furthest values within 1.5x the inter-quartile range. All pairwise group comparisons (both between and within prediction types) are significant ($p < 2.5 \times 10^{-28}$, two-sided Dunn's test for multiple pairwise comparisons, corrected for multiple testing by Holm's method; see 'Methods'). **b** Receiver operating characteristic (ROC) curves and area under the curve (AUC) for discrimination between ClinVar and gnomAD mutations using monomer and full |ΔΔG| values. Source data are provided as a Source Data file.

Our previous study on the identification of pathogenic mutations with protein stability predictors considered only monomeric proteins[18]. As the dataset used in our current study was derived from both monomeric and complex structures, we first investigated the impact of using full structures. For each mutation, we calculated ΔΔG for both the protein monomer alone, and for the full biological assembly containing all molecules. Monomer |ΔΔG| represents the effect on the stability of a monomeric protein or isolated subunit, while full |ΔΔG| includes both intra- and intermolecular interactions. For the structures of monomeric proteins, and for mutations not close to intermolecular interfaces, the full and monomer |ΔΔG| values will be identical or nearly identical.

Figure 1a shows that using the full structures in the FoldX calculations increased the average extent of observed stability perturbations of pathogenic mutations by almost 15%, from 3.39 to 3.89 kcal mol$^{-1}$, compared to only using the monomeric subunits. To see how this affects the identification of disease mutations, we performed receiver operating characteristic (ROC) analysis to assess discrimination between the ClinVar and gnomAD mutations. The area under the curve (AUC), which corresponds to the probability of a randomly chosen disease mutation being assigned a higher-ranking |ΔΔG| value than a random gnomAD variant[26], was used as a quantitative classification performance metric. As is evident from Fig. 1b, the curve, derived from full |ΔΔG| values, resides higher than the performance curve of monomer prediction values over the entire threshold range. Using full |ΔΔG| values results in an AUC of 0.677, which is significantly higher ($p = 4.3 \times 10^{-71}$) than the AUC of 0.652 when only using monomeric structures. While this is unsurprising, given the common role of intermolecular interactions in human genetic disease[27,28], it emphasises the importance of considering full protein complex structures, when available.

**Recessive mutations are more structurally perturbing than dominant mutations.** Next, we investigated mode of inheritance, which is closely related to molecular disease mechanism. Autosomal recessive (AR) disorders are overwhelmingly associated with LOF, whereas autosomal dominant (AD) disorders can have different underlying molecular mechanisms[23]. While some dominant mutations will cause disease via LOF (i.e., haploinsufficiency), many will be DN or GOF. Thus, we expect that differences in the structural effects of recessive vs dominant missense mutations should be reflective of the differences between LOF and non-LOF mechanisms.

Disease inheritance annotations for genes were obtained from OMIM[29]. To allow a simplified analysis at the gene level, we only investigated genes with either autosomal recessive (726 genes) or autosomal dominant (535 genes) inheritance, excluding those with mixed inheritance. Figure 2a compares the monomer and full |ΔΔG| values for AD and AR disease mutations, and the putatively benign gnomAD variants. While all groups exhibit a high degree of heterogeneity, AR mutations are significantly more perturbing, with a mean difference of 1.1 kcal mol$^{-1}$ for monomer |ΔΔG| and 0.9 kcal mol$^{-1}$ for full |ΔΔG| compared to AD mutations. Interestingly, AD mutations are approximately intermediate compared to gnomAD and AR variants.

The differences in perturbation magnitude across the different mutation groups can be partially explained by their enrichment in different spatial locations (Fig. 2b). It is well known that pathogenic mutations are common in protein interiors and at interfaces, where they can act via protein destabilisation or disruption of interactions, while they are underrepresented on protein surfaces[27,28,30–32]. We find that AR mutations are most enriched in protein interiors (58%), while only 15% occur at the protein surface. In contrast, 43% of the gnomAD variants occur on the surface and 38% in the interior. AD mutations appear intermediate between AR and gnomAD, with 43% in the interior and 26% on the surface. Interestingly, however, the AD group shows the highest prevalence of variants at protein interfaces (31%), compared to 27% for AR and 20% for gnomAD, hinting at the importance of intermolecular interactions for understanding alternate molecular disease mechanisms. Importantly, in Supplementary Fig. 2 we show that the differences in perturbation magnitudes are still observed when controlling for interior, interface and surface locations.

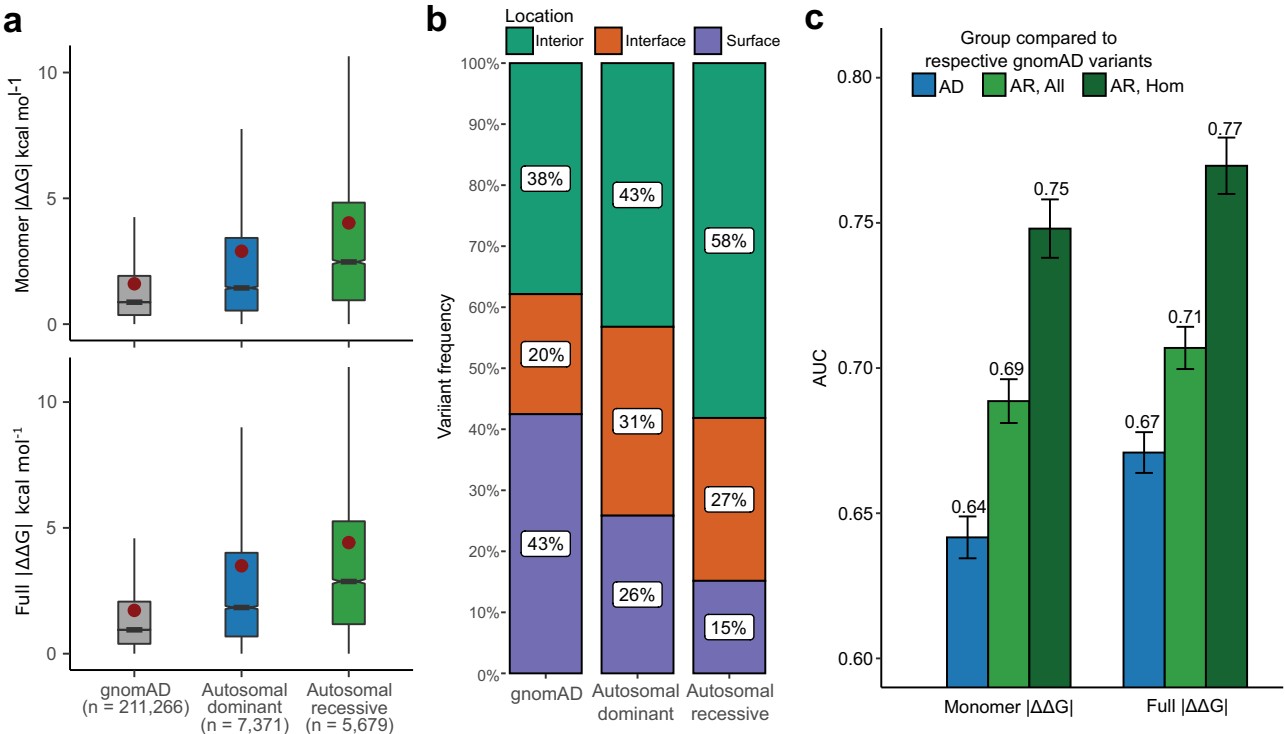

**Fig. 2 Autosomal dominant mutations are milder than autosomal recessive mutations. a** Comparison of predicted $|\Delta\Delta G|$ values for gnomAD variants and pathogenic ClinVar mutations from autosomal dominant (AD) and autosomal recessive (AR) genes. Boxes denote data within 25th and 75th percentiles, and contain median (middle line) and mean (red dot) value notations. Whiskers extend from the box to furthest values within 1.5x the inter-quartile range. All pairwise comparisons between groups are significant ($p < 6.4 \times 10^{-72}$, two-sided Holm-corrected Dunn's test). **b** Proportions of gnomAD, AD and AR mutations occurring in protein interiors, surfaces and interfaces. All pairwise comparisons are significant (Chi-square test). **c** AUC values calculated from ROC curves for discriminating between pathogenic ClinVar mutations from AD or AR genes, and putatively benign gnomAD variants. 'AR, All' considers all gnomAD variants from the AR genes, while 'AR, Hom' only includes those variants that have been observed in a homozygous state in gnomAD at least once. Error bars denote 95% confidence intervals. Source data are provided as a Source Data file.

As the ability to identify pathogenic mutations also depends upon the properties of the benign variants from which they must be discriminated, we further compared the $|\Delta\Delta G|$ values of gnomAD variants from AD *vs* AR disease genes. The properties of gnomAD variants from AR genes are expected to be quite different compared to those from AD genes, as AR genes will, by definition, tend to be more tolerant of heterozygous damaging mutations. Therefore, we selected a subset of variants that have been observed in a homozygous state at least once in gnomAD, with the expectation that these should be more reflective of truly benign variants (although at the cost of being a much smaller dataset). Interestingly, we find in Supplementary Fig. 3 that, while the gnomAD variants in AR genes are collectively much more damaging than those from AD genes, when we consider the homozygous subset of variants from AR genes, they are very similar to AD gene mutations.

Finally, we assessed whether the inheritance mode of variants affects the performance of disease variant identification using predicted stability effects. We observe considerably increased performance in the identification of pathogenic missense mutations from AR genes compared to AD genes as measured by ROC AUC (0.71 vs. 0.67 for full $|\Delta\Delta G|$, AR and AD respectively), which is even greater when the homozygous set of putatively benign AR gnomAD variants are used (0.77; Fig. 2c). This suggests that predicted changes in protein stability are more useful for the identification of recessive mutations, due to the fact that recessive disorders are much more likely to be associated with LOF.

**Gain-of-function and dominant-negative mutations have much milder effects on protein structure than loss-of-function mutations.** To compare the effects of mutations associated with different molecular disease mechanisms, we attempted to classify AD disease genes into those associated with haploinsufficiency (HI), DN effects, or GOF, using a combination of keyword searches, manual curation of OMIM[29] entries, and ClinGen[33] annotations (see 'Methods'). While this approach is necessarily imperfect, in that it assumes that all disease mutations from the same gene will be associated with the same mechanism, it represents the most feasible strategy we currently have available for investigating the general properties of these different types of mutations on a larger scale. For the sake of flow and conciseness, from this point in the text we will be referring to the variants from classified genes directly by the associated mechanism ('DN mechanism variants' and not 'variants from genes associated with DN disease').

We first explored predicted stability effects using monomer $|\Delta\Delta G|$ values (Fig. 3a, left). Interestingly, a clear difference emerges between the loss-of-function (HI) and non-LOF (DN and GOF) mutations, with the HI mutations being far more disruptive to protein structure. The HI mutations show a nearly identical distribution to the AR mutations, with monomer $|\Delta\Delta G|$ means of 4.18 and 4.02 kcal mol$^{-1}$, respectively, and no statistically significant difference. In contrast, the DN and GOF mutations are much milder, with monomer $|\Delta\Delta G|$ means of 2.66 and 2.42 kcal mol$^{-1}$, respectively, and also showing no statistically significant differences from each other. In fact, although the

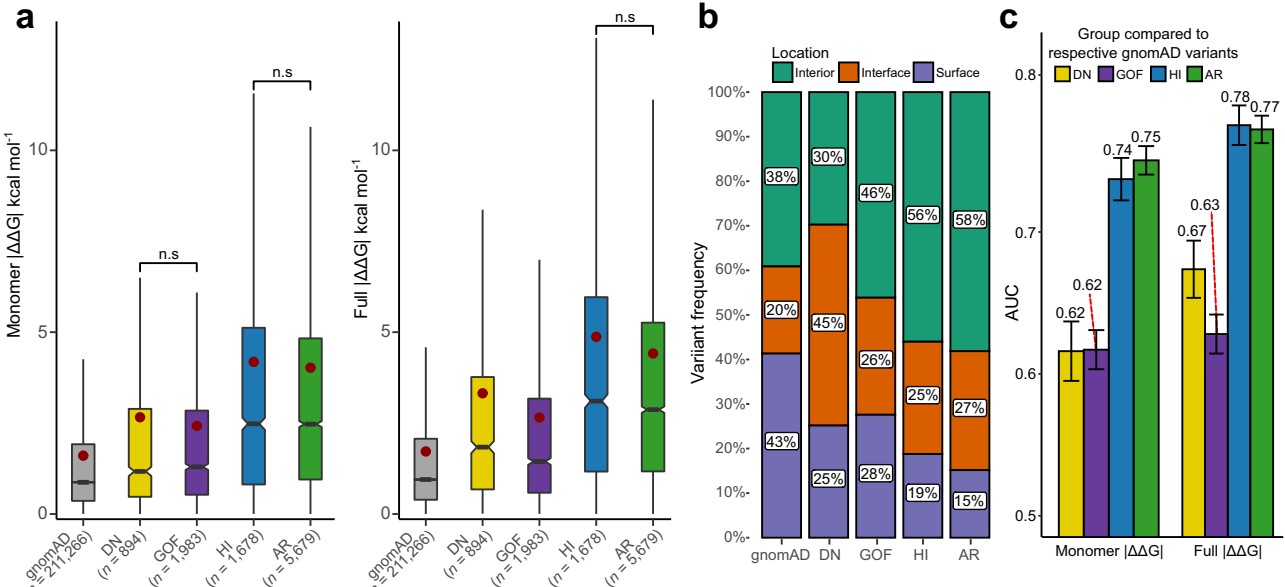

**Fig. 3 Gain-of-function and dominant-negative mutations are milder than loss-of-function mutations. a** Comparison of predicted $|\Delta\Delta G|$ values for gnomAD variants and pathogenic ClinVar mutations from AD genes classified as dominant negative (DN), gain of function (GOF) and haploinsufficient (HI), and pathogenic ClinVar mutations from AR genes. Boxes denote data within 25th and 75th percentiles, and contain median (middle line) and mean (red dot) value notations. Whiskers extend from the box to furthest values within 1.5x the inter-quartile range. Pairwise group comparisons are significant ($p < 0.001$, two-sided Holm-corrected Dunn's test) unless specified (n.s., $p$-values in order from left to right: 0.786, 0.321, 0.075). **b** Proportions of different types of mutations occurring in protein interiors, surfaces and interfaces. All pairwise proportion comparisons via Chi-square test are significant, with the lowest effect size being noted for HI vs AR (Cramer's $V = 0.04$, compared to 0.20 and 0.26 for DN vs. HI, and DN vs. AR, respectively). **c** AUC values calculated from ROC curves for discriminating between different types of pathogenic ClinVar mutations, and putatively benign gnomAD variants. Only homozygous gnomAD variants were included for the AR analysis. Error bars denote 95% confidence intervals. Source data are provided as a Source Data file.

DN and GOF mutations are considerably more perturbative than the putatively benign gnomAD variants, they are closer to those in magnitude than they are to the LOF mutations.

When intermolecular interactions from full protein structures are also taken into consideration, using full $|\Delta\Delta G|$ values (Fig. 3a, right), a similar overall pattern is observed, with LOF mutations being the most damaging, and DN and GOF mutations being intermediate between LOF and gnomAD. However, DN mutations show a large rise in predicted $|\Delta\Delta G|$ compared to using only monomeric structures, increasing from 2.66 to 3.32 kcal mol$^{-1}$, while GOF mutations show only a small increase, from 2.42 to 2.65 kcal mol$^{-1}$, and the overall difference between DN and GOF mutations becomes highly significant.

Figure 3b demonstrates that the DN mutations are far more likely to affect intermolecular interactions, with 45% of mutations occurring at the interface. In contrast, GOF mutations show little interface enrichment, thus explaining the significant difference between DN and GOF mutations when considering full $|\Delta\Delta G|$ values. Of course, this result is probably influenced by the fact that DN mutations are more likely to be found in protein complexes[22], with our dataset showing a significant enrichment of DN-associated mutations in complexes vs monomers ($p = 1.581 \times 10^{-10}$, Fisher's exact test), compared to the complex-monomer proportion of all other disease variants.

Using ROC analysis, we tested the ability of the predicted stability perturbations to distinguish between pathogenic and putatively benign gnomAD variants across all the different molecular disease mechanism groups (Fig. 3c). Most strikingly, we see that LOF mutations (HI and AR) are predicted far better than DN and GOF mutations, with AUCs 0.1–0.15 higher for LOF vs non-LOF groups. We also note a 0.05 AUC improvement in the prediction of DN mutations when considering intermolecular

effects through full $|\Delta\Delta G|$ values, consistent with our observations above, although this is still well below the performance observed for LOF mutations.

We previously demonstrated that optimal performance of stability predictors for the identification of pathogenic mutations is subject to significant gene and predictor-specific heterogeneity[18]. While FoldX $|\Delta\Delta G|$ values showed an optimal threshold for distinguishing between pathogenic and benign close to 1.5 kcal mol$^{-1}$, a value that has been previously utilised in several studies on variant stability perturbation[34–36], we observed that on a per-gene level the optimal thresholds varied considerably. Therefore, we explored whether knowledge of the underlying molecular disease mechanism could inform us of a more optimal threshold to use for variant prioritisation. Supplementary Table 1 lists the optimal FoldX $|\Delta\Delta G|$ thresholds for gene groups based on our molecular mechanism annotations. We observe considerable differences in optimal thresholds, ranging from 1.16 kcal mol$^{-1}$ for GOF mutations to 1.59 kcal mol$^{-1}$ for HI mutations when using full $|\Delta\Delta G|$. Interestingly, the two LOF group thresholds correspond quite closely to the previously used 1.5 kcal mol$^{-1}$. However, our findings signify that using this single threshold in a practical setting would lead to considerable underprediction of non-LOF disease variants. It is currently unlikely that practical prediction performance can be dramatically improved by mechanism-specific threshold choices, but awareness of the unique protein-and variant-level features underlying non-LOF variants could reveal future avenues for protein-specific prediction approaches.

**There are significant functional class prevalence differences across disease inheritance and molecular mechanism groups.** Given that protein structure is intrinsically tied to biological

function, we wondered whether our results could in part be explained by different molecular disease mechanisms being related to different functional contexts. It has been previously shown that taking into account protein functional class annotations can increase accuracy in distinguishing between disease and benign variants[37]. Expanding upon the dataset derived by Iqbal et al.[37] through manual and PANTHER[38] annotation, we derived a 25-class functional protein annotation for our genes, with a single gene being allowed to be associated with more than one functional class. Supplementary Fig. 4 demonstrates the functional class label prevalence differences for gene groups associated with distinct inheritance and molecular disease mechanisms, with statistically significant class proportion differences observed for all comparisons except between DN and GOF disease genes (Holm-corrected Chi-square test $P = 0.15$; remaining comparison $P < 4.12 \times 10^{-3}$). There are major functional class differences between AD and AR genes, particularly with transporters and the overlapping categories of transcription factors and nucleic acid binding proteins being enriched in dominant disease, and various enzymatic functions being primarily associated with recessive disease. However, when we control for molecular disease mechanism, we can see that there are significant functional class prevalence differences: the dominant enrichment in transcription factors and nucleic acid binding proteins is overwhelmingly driven by HI genes; GOF disease is more associated with signaling molecules, some enzyme classes and transporters; and DN effects tend to occur more predominantly in receptors, transcription factors and transporters.

To assess whether the observed general stability change trends may in fact be driven by the underlying biological functions, we compared the predicted full $|\Delta\Delta G|$ value differences between mechanisms for each functional class. We only explored functional classes that had at least 20 pathogenic variants associated with each of the four disease mechanisms groups (DN, GOF, HI, AR). As we see in Supplementary Fig. 5, the same general tendency for LOF variants to be more damaging than non-LOF variants is observed across most groups. Notably, HI and AR variants are significantly more damaging than GOF variants in most groups. However, in kinases, signalling molecules, receptors and transferases, DN variants are actually the most damaging, although this could possibly be related to small sample sizes. This suggests that the precise molecular mechanisms underlying DN mutations could show some tendency to vary related to functional context. For example, while a DN effect can be caused by non-destabilising mutants that can incorporate into and "poison" a protein complex, there are also DN mutations that disrupt interactions, resulting in a "competitive" DN effect[39].

**Independent variant-level GOF and LOF dataset supports the observed stability effect trends.** To validate our generalised variant disease mechanism annotation approach, we took advantage of a recently published dataset containing variant-level GOF *vs* LOF mechanism assignments from Bayrak et al.[40], derived through a natural language processing model applied to available literature. Using their HGMD[41] missense mutation annotations as a foundation, we derived a structural FoldX score dataset based on predicted AlphaFold[42] monomer models (see 'Methods'), as many of the proteins in this dataset lacked experimentally determined structures. The external dataset consisted of variants from 361 OMIM 'AD', 353 'AR' and 83 mixed inheritance 'ADAR' genes (containing both AD and AR disease variants), with only 76 genes from the total harbouring both GOF and LOF variants (Supplementary Fig. 1).

As we demonstrate in Supplementary Fig. 6a, HGMD GOF variants are significantly milder than LOF variants in terms of $|\Delta\Delta G|$ values in all scenarios we explored: considering the full dataset; considering only variants from mixed inheritance genes; and considering only genes with both GOF and LOF variants. Furthermore, for those genes with both GOF and LOF variants, we find that the GOF variants have a significant tendency to be milder than the LOF variants from the same gene (Supplementary Fig. 6b). These analyses also validate our gene-level classification approach, as, interestingly, even when annotated at variant-level, a large majority of the genes contain either GOF or LOF disease variants, exclusively. Furthermore, the observed differences between GOF and LOF variants should be even more pronounced if controlling for DN variants, as natural language processing model used in the Bayrak et al. study did not distinguish dominant-negative variants separately; thus, the LOF variant class is likely to contain some proportion of DN variants.

Using the AlphaFold models we also explored the per-residue modelling quality metric, pLDDT, which has been shown to also be highly predictive of structural disorder[43]. We compared pLDDT between the HGMD GOF and LOF variants, as well as our dataset of ClinVar variants split into four mechanism classes. As we derived our annotations for ClinVar variants at the gene-level, we present the results as per-gene disease variant pLDDT means, to account for uneven variant sample sizes between the genes. While the shift is not drastic, Supplementary Fig. 7a shows that HGMD GOF variants occur at positions characterised by significantly lower pLDDT, indicating the regions are likely more disordered. This makes sense for GOF variants in terms of the observed milder $\Delta\Delta G$ values, as lower residue density in a less-ordered region would lead to fewer unfavourable energetic interactions. Additionally, fewer mutations are needed in intrinsically disordered regions to give rise to new interactions, compared to structured domains, increasing the likelihood of disease via gain-of-function mechanisms in such regions[44]. Interestingly, we also observe a significantly lower pLDDT in the ClinVar GOF genes, compared to the LOF groups (HI and AR), although the distinction from DN genes is not significant (Supplementary Fig. 7b). However, looking at the overall disease variant density according to pLDDT, we can see that that disease variants in disordered regions (pLDDT < 50) represent the minority of currently known pathogenic variants, for both GOF and LOF disease (Supplementary Fig. 8).

Finally, to validate the disease variant identification performance differences between the distinct mechanisms, we subjected the external HGMD dataset to a classification strategy similar to our gene-level ClinVar annotation, using our own haploinsufficiency assignments to split the dominant LOF mutations into HI and 'Other LOF' groups, with the 'Other LOF' group expected to be enriched in DN mutations (see 'Methods'). Reassuringly, we find that the four-group HGMD variant performance closely resembles what we demonstrated using our ClinVar dataset (Supplementary Fig. 9). The results are not surprising, as the vast majority of genes in the HGMD contain disease variants characterised by a single disease mechanism class, which also suggests our gene-level annotation approach may not be strongly prone to bias.

**Stabilising mutations may be important for disease in gain-of-function and particular loss-of-function contexts.** We also hypothesised that, in addition to being milder in magnitude, pathogenic GOF mutations might have a tendency to increase protein stability, e.g., by stabilising activated states. Similarly, DN mutations could conceivably be associated with increased

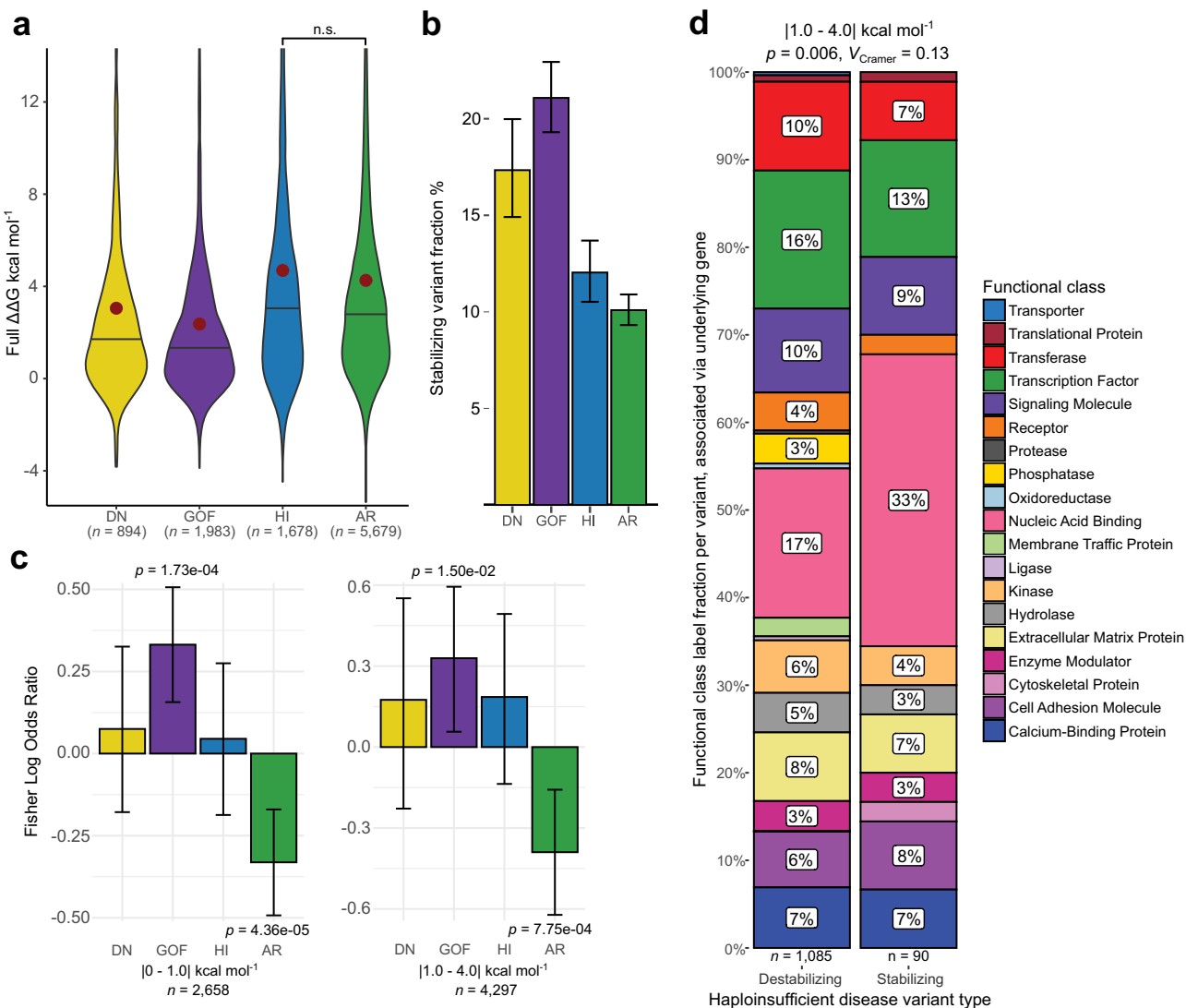

**Fig. 4 Stabilising mutations may be important for disease in gain-of-function and particular functional class LOF contexts. a** ΔΔG distributions (not absolute values) for pathogenic mutations, associated with different molecular mechanisms. Violin middle line shows median, red dot represents the mean. Pairwise group comparisons are significant ($p < 3.29 \times 10^{-4}$, two-sided Holm-corrected Dunn's test) unless specified (n.s., $p = 0.107$). **b** Fraction of pathogenic mutations associated with different molecular mechanisms with negative (stabilising) ΔΔG values. Error bars denote 95% confidence intervals. **c** Fisher log odds enrichments of stabilising mutations within each class from among all disease mutations. Error bars represent 95% confidence intervals. **d** Destabilising and stabilising variant prevalence differences in HI genes associated with particular functional classes. Sample sizes denote variant number, found in a gene associated with the particular function. Chi-square test p-value and effect size are shown. Source data are provided as a Source Data file.

rigidification, allowing a mutant protein to 'poison' a complex containing a mixture of mutant and wild-type subunits. Indeed, a number of pathogenic mutations that increase protein stability have been noted in the literature[45–47]. To address this, we need to consider the actual ΔΔG values, including sign. In Fig. 4a, we plot the distribution for full ΔΔG values different types of pathogenic mutations. From this, we can see that, while the large majority of mutations in each class are predicted to have a net destabilising effect (ΔΔG > 0), there are still many stabilising mutations. Figure 4b shows the fraction of stabilising mutations (ΔΔG < 0) for each group. This shows that predicted stabilising effects are most common in GOF mutations (21%), followed by DN (17%). In contrast, LOF mutations have a much lower tendency to be stabilising (12% for HI and 10% for AR).

While the above analysis shows that non-LOF mutations are more likely to have predicted stabilising effects, it is possible that this is simply due to the fact that they tend to be of lower magnitude. If one assumes a typical error of ~1 kcal mol$^{-1}$ for a FoldX ΔΔG prediction[48], then we can expect that many of the low magnitude ΔΔG values might appear to be stabilising due to this margin of error. We attempted to control for this by considering predicted stabilising vs destabilising effects in two groups: those with $|\Delta\Delta G| < 1$ kcal mol$^{-1}$, and those with $|\Delta\Delta G|$ from 1-4 kcal mol$^{-1}$. We then calculated Fisher log odds enrichment of stabilising mutations within each class from among all disease mutations annotated with a mechanism. As illustrated in Fig. 4c, the tendency for predicted stabilisation to be most enriched among GOF mutations is observed in both groups ($p = 1.73 \times 10^{-4}$ and 0.015 for $|\Delta\Delta G| < 1$ kcal mol$^{-1}$ and 1–4 kcal mol$^{-1}$ groups, respectively). In contrast, destabilisation is most enriched among AR mutations ($p = 4.36 \times 10^{-5}$ and $7.75 \times 10^{-4}$). Interestingly, stabilisation appears to be more common in HI than AR mutations, which is difficult to explain, considering that we would expect both to act via similar LOF mechanisms. While both of these mechanism groups were previously shown to be similar in terms of both overall

predicted variant perturbation magnitude and structural variant location, HI seems to show a higher tendency of eliciting pathogenicity through stabilising, or rigidifying, the structure or interaction.

One explanation for this may be that HI genes differ from the AR group in terms of specific functional class prevalence. As previously seen in Supplementary Fig. 4, mutant transcription factors are among those proteins more susceptible to cause disease through haploinsufficiency, and the importance of their conformational flexibility in divergent sequence recognition has been previously explored[49,50]. Increasing transcription factor stability may rigidify them and reduce their conformational flexibility, in turn causing a loss of function. To test this hypothesis, we split up HI gene variants in the $|1-4|$ kcal mol$^{-1}$ bin, according to their predicted effect type, either destabilising or stabilising, and observed the differences in relative prevalence of functional protein class labels of their associated genes (Fig. 4d). While stabilising variants were slightly less likely to be associated with transcription factors than destabilising variants (13% vs. 16%), the closely related nucleic acid binding category was almost twice as likely to be associated with stabilising mutations (33% vs. 17%). This result could signify that stabilising variants are important to understanding disease that stems from dysfunction of nucleic acid binding as a whole in haploinsufficiency contexts. Additionally, our analysis demonstrates how knowledge of protein functional classes and underlying molecular disease contexts can potentially be valuable as synergistic features for variant identification. We also repeated the analysis for remaining mechanism groups; however, they did not show statistically significant functional gene class differences between stabilising and destabilising variants.

**Nearly all variant effect predictors underperform on non-loss-of-function mutations**. Given that DN and GOF mutations tend to be mild at a protein structural level, and are thus poorly identified using FoldX$|\Delta\Delta G|$ values, we wondered whether this might also have similar implications for the performance of other computational VEPs. Initially, our hypothesis was that, while predictors that incorporate protein structural information might perform worse on non-LOF mutations, those based primarily on evolutionary conservation should be relatively insensitive to differences in molecular disease mechanism. Therefore, we tested a diverse set of 45 VEPs for their performance on different types of disease mutations.

Supplementary Table 2 outlines the different VEPs used, which we classified into several groups. First, we included those based purely on amino acid sequence conservation. Multi-feature methods are those that rely on multiple different features, although they all also include sequence conservation. Metapredictors are those VEPs that derive predictions based on the outputs of two or more other predictors. We also considered several simple amino acid substitution matrices. Finally, we considered some methods based purely on DNA-level features, primarily nucleotide conservation. It is important to note that a majority of the VEPs in our multi-feature and metapredictor groups were derived in a supervised fashion, which makes them susceptible to circularity due to dataset overlaps with training data[7,51]. While this prevents us from being able to make fair judgements on absolute performance of different predictors, as some VEPs are likely to have been trained on some of the mutations in our dataset, we are more interested in the relative performance within predictors against different molecular disease mechanism groups.

Figure 5 shows the ROC AUC values for discriminating between pathogenic ClinVar and putatively benign gnomAD

missense variants across all predictors. Remarkably, we observe nearly universal underperformance on non-LOF mutations. For the large majority of protein-level predictors, the LOF mutations (HI and AR) show higher AUC values than the non-LOF (DN and GOF) mutations. Furthermore, when we subject our four-class HGMD dataset based on variant-level GOF vs. LOF annotation to the same analysis, we observe very similar results (Supplementary Fig. 10), despite the fact that only 329 of the 714 HGMD genes overlap with our 1261 gene ClinVar dataset.

How can we explain this consistent underperformance against non-LOF mutations? One clue comes from examination of the substitution matrices. These are very simple models, derived from physicochemical properties of amino acids, or patterns of evolutionary substitutions, that will always give the same value for the same type of amino acid substitution. Effectively, they represent how different the mutant amino acid residue is from the wild type. Across all substitution matrices, HI mutations are predicted better than DN and GOF mutations, which simply tells us that the non-LOF mutations tend to involve more similar amino acid substitutions than LOF mutations. Thus, despite the fact that these models do not use protein structural information, they are still reflecting the milder nature of the pathogenic non-LOF substitutions.

In principle, the substitution matrix results could also explain the performance of some other predictors based on amino acid sequence conservation that also incorporate a substitution matrix (e.g. SIFT[52]). Notably, however, the underperformance of non-LOF mutations is also observed for DeepSequence[53], which is based purely on multiple sequence alignments of the protein of interest and does not explicitly use any external substitution scores for variant evaluation, suggesting that amino acid residues associated with DN or GOF mutations genuinely tend to show weaker evolutionary conservation than those associated with LOF mutations.

It is also interesting to note the performance differences of the nucleotide-level prediction methods between the ClinVar and HGMD datasets (Fig. 5 vs. Supplementary Fig. 10). For the HGMD variants, nucleotide-level methods perform worse on all dominant mutation groups, while for ClinVar variants, only DN and HI genes show underperformance. One explanation is that the HGMD dataset, being based on AlphaFold models, does include intrinsically disordered proteins or regions, which, if not involved in a particular function, are less likely to be conserved[54]. As we showed earlier, GOF variants were more likely to occur at positions of lower AlphaFold pLDDT, a metric intrinsically related to the degree of structural order. The PDB-based ClinVar dataset, lacking disordered regions, may be enriched in GOF variants at more conserved positions, resulting in higher GOF variant identification performance by conservation-based methods.

Overall, the tendency for LOF mutations to be predicted better than non-LOF mutations is very clear. However, the differences are relatively small between DN and GOF mutations across most predictors. Interestingly, the DN mutations are better predicted across all substitution matrices, suggesting that DN mutations tend to involve more perturbative amino acid substitutions. In contrast, GOF mutations tend to be better predicted by methods based purely on sequence conservation. The two competing factors - DN mutations being somewhat more perturbative, and GOF mutations occurring at more conserved positions – appear to effectively cancel each other out across most VEPs that utilise both sources of information, resulting in very similar predictions for the two groups. Comparison of conservation and amino acid substitutions may hold some value in discriminating between DN and GOF mutations.

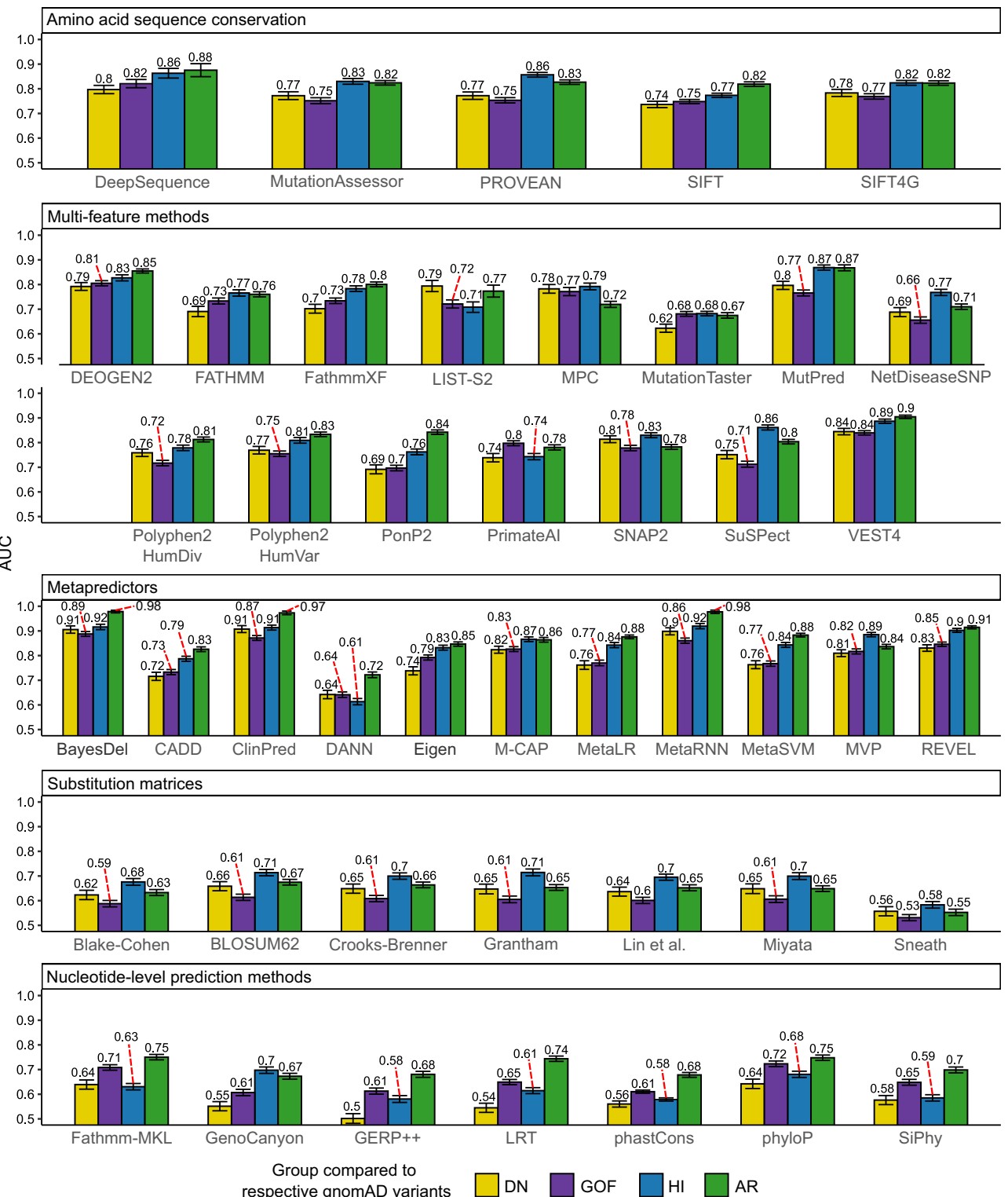

**Fig. 5 Dominant-negative and gain-of-function mutations are less well identified by nearly all computational variant effect predictors than loss-of-function mutations.** AUC values calculated from ROC curves for discriminating between different types of pathogenic ClinVar mutations and putatively benign gnomAD variants, using the outputs of different computational variant effect predictors. Only homozygous gnomAD variants were included for the AR analysis. Error bars denote 95% confidence intervals. Source data are provided as a Source Data file.

**Gain-of-function and dominant-negative mutations tend to cluster in three-dimensional space.** Finally, since non-LOF mutations tend to be poorly predicted by existing computational approaches, we wondered whether there is other information that could be used to better predict them. We hypothesised that DN

and GOF missense mutations should tend to cluster within specific regions, while LOF missense mutations should be more evenly spread throughout a protein. The reason for this is simple: destabilising mutations should be able to occur throughout a protein sequence, at least within its folded domains, while the

more specific effects associated with DN and GOF mutations may be more localised to particular regions. Previously, there has been some evidence for this. For example, a recent study found that genes with more spatially clustered disease variants tend to be associated with mechanisms other than haploinsufficiency[55]. Other studies have shown dominant GOF mutations exhibit more focal, shorter distance clustering at sites of functional potential, compared to recessive LOF variants, which occur more uniformly throughout structures[56,57]. Another study of cancer-associated variation found that GOF mutations in oncogenes tend to be more clustered compared to LOF mutations in tumour suppressors and other targets[58].

To address this in more detail, we defined a simple metric for disease mutation clustering, based on the proximity of each protein residue to a known disease mutation at another residue within the same polypeptide chain (see 'Methods'). The final clustering metric is presented as a ratio for each protein, termed the EDC ('Extent of Disease Clustering'), whereby a value of greater than one indicates that residues where disease mutations occur tend to cluster together within the three-dimensional structure of the protein, whereas a value of one would be expected if the sites of disease mutations were randomly distributed. The rare examples of proteins with clustering ratios less than one indicate cases where the sites of disease variants are more evenly distributed throughout the protein, but these are likely to represent chance occurrences due to the nature of the metric rather than meaningful "anti-clustering".

In Fig. 6a, we compare the distributions of clustering values for AR vs AD disease genes, under the assumption that AR genes should be primarily associated with LOF mechanisms, while AD genes can be associated with LOF or non-LOF mechanisms. Remarkably, we observe a very strong, highly significant tendency for disease mutations in AD genes to be more clustered than those in AR genes. Furthermore, if we consider a variation of the metric derived using putatively benign gnomAD variants for the same genes, rather than pathogenic variants, we observe a much lower extent of clustering. In Fig. 6b, we compare the distributions of clustering values for AD genes associated with different molecular disease mechanisms. Consistent with our previous observations, we find that DN and GOF mutations tend to be significantly more clustered than HI mutations. Together, these results strongly support the idea that non-LOF disease mutations tend to cluster together more than LOF mutations. Interestingly, however, there is no significant difference between clustering of DN and GOF mutations. Thus, the utilisation of information about mutation clustering in three-dimensional space shows promise for improving predictions of pathogenic non-LOF mutations.

To exemplify the utility of this finding, we considered two genes on the opposite ends of the EDC spectrum, ADSL and KCTD1, which are associated with dominant and recessive disease, respectively (Fig. 6c). LOF mutations in ADSL cause adenylosuccinate lyase deficiency[59], and the protein is characterized by a notably low EDC value of 0.89. This indicates strong disease variant dispersal throughout the structure. KCTD1, on the other hand, demonstrates a high degree of disease variant clustering (EDC = 1.90). However, KCTD1 was not annotated with a molecular disease mechanism in our pipeline, as it did not have a ClinGen dosage sensitivity entry or references to dominant-negative or gain-of-function mechanisms in OMIM. Interestingly, exploration of the literature revealed numerous cases of the mechanism underlying KCTD1 disease being described as dominant negative, in reference to causing the scalp-ear-nipple syndrome. Thus, EDC can in practice serve as a tool to identify disease genes with mutations likely to act via non-loss-of-function mechanisms. Certainly, we believe there is much

scope in the future to consider alternate approaches for quantifying mutation clustering, and considering intermolecular distances within protein complexes, which could lead to significantly improved disease variant identification.

## Discussion

Our first observation in this study was the importance of considering full protein complex structures, if available, when using stability predictors to investigate potential mutation pathogenicity. This is unsurprising, as many pathogenic mutations are known to occur at protein interfaces[32], and the impact of these will be missed when considering only monomeric structures. Thus, the effects of a mutation on a protein complex structure should be tested if possible. While there will be many cases where no structure of a biologically relevant complex is available, advances in protein modelling and intermolecular docking may be helpful in the future.

Next, we found a striking association between predicted changes in protein stability, inheritance and molecular disease mechanisms, which were independent of the distinct functional class enrichments in those groups. LOF mutations associated with AR and HI genetic disorders can be distinguished quite well using changes in protein stability, but dominant GOF and DN mutations tend to have mild effects at a protein structural level. While there has been limited evidence of this in the past for DN mutations in transmembrane proteins[14], this is the first time this has been assessed on such a large scale for both DN and GOF mutations.

Given the above observations, it is essential to emphasise that the primary benefit of stability predictors with respect to disease mutations is in understanding molecular mechanisms. When a mutation is predicted to destabilise a protein or disrupt an interaction, this gives us a plausible molecular disease mechanism. However, that a mutation is structurally mild should not be taken as strong evidence that it is likely to be benign, unless all other known pathogenic mutations are known to be destabilising. There is a tendency in the literature when reporting new disease mutations to only include predicted changes in stability if they are high. We suggest that even mild changes in predicted or observed stability should be reported, as this can provide clues as to the potential molecular mechanisms. With additional further understanding of molecular disease mechanisms in the context of other features, like protein functional classes, we may be able to individualise stability score thresholds for precision variant identification.

One limitation of our study arises from our strategy for annotating HI vs DN vs GOF mutations. We are reliant on the descriptions of mutations that have been compiled into the OMIM database, which can be somewhat qualitative. In fact, it is often very difficult to distinguish between DN and GOF mutations experimentally, and thus there is a large set of mutations we excluded from our analyses because OMIM described them as potentially being either DN or GOF, but could not distinguish between the two. Moreover, for simplicity, we assume that all pathogenic missense mutations in the same gene are associated with the same molecular mechanism, which is clearly not true for all genes. Importantly, however, we were able to reproduce our results using a recent variant-level dataset, annotated with GOF and LOF mechanism assignments[40], thus supporting the validity of our gene-level approach. However, currently available variant-level datasets do not cover DN variants on a large-scale. It is highly likely that, if we had more accurate, mutation-level classifications of molecular mechanisms of disease mutations, the trends we have observed here would be even more pronounced.

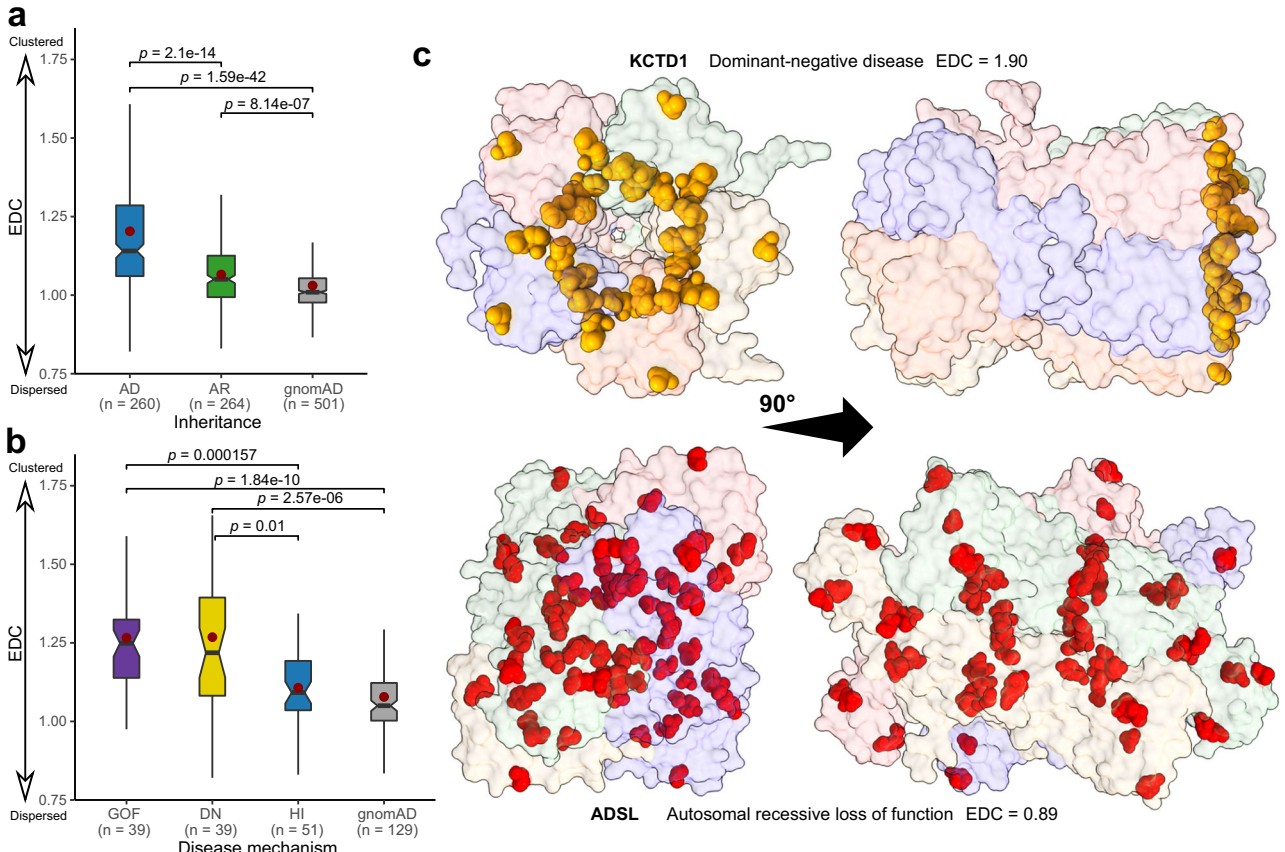

**Fig. 6 Pathogenic mutations in genes associated with dominant-negative and gain-of-function mechanisms are significantly more clustered in space than those associated with loss-of-function mechanisms.** Sample sizes denote protein number. Boxes denote data within 25th and 75th percentiles, and contain median (middle line) and mean (red dot) value notations. Whiskers extend from the box to furthest values within 1.5x the inter-quartile range. **a** Distribution of Extent of Disease Clustering (EDC) values for autosomal recessive (AR) compared to autosomal dominant (AD) disease genes. EDC values calculated with gnomAD variants from the same set of genes are shown for comparison. p-values were calculated with the two-sided Holm-corrected Dunn's test. EDC values were only calculated for protein structures with pathogenic/gnomAD variants at least 10 different residues, which means that the number of proteins in the gnomAD set is slightly smaller than the sum of the AD and AR groups. **b** Distribution of EDC values for AD disease genes associated with gain-of-function (GOF), dominant-negative (DN) and haploinsufficient (HI) mechanisms. **c** Demonstration of the variant distribution differences between non-LOF (KCTD1) and LOF (ADSL) disease proteins, and their strikingly different EDC values. Source data are provided as a Source Data file.

While we have shown that DN and GOF mutations tend to have mild effects on protein stability compared to LOF mutations, the actual specific mechanistic details underlying most non-LOF mutations remain indeterminate. We know that the DN effect tends to be associated with intermolecular interactions and the assembly of proteins into complexes[22,60,61], and this is supported by our observation of an enrichment of DN mutations at interface positions. However, our finding that including intermolecular interactions and using the full |ΔΔG| values significantly improve the identification of pathogenic DN mutations is in a way para-doxical, as it implies that DN mutations tend to perturb inter-actions. A mutation that completely disrupts assembly of a protein complex would not be compatible with a DN effect, as there would be no way for the mutant protein to affect the wild-type protein. However, it is possible that DN mutations often cause milder perturbations, not entirely blocking protein complex assembly, but sufficient to affect the function of a full complex. Alternatively, by 'loosening' interactions in the complex, DN mutations may cause all subunits, including wild type, to be mislocalised or degraded, as in the example of ALDH2[22,62]. Conversely, the observed increase in perturbation could in some cases stem from stabilising mutations involved in intermolecular interactions, inactivating complexes through rigidification of

conformational dynamics, competing for and sequestering lim-iting substrates or targets in non-productive complexes[61,63]. As for GOF mutations, they are likely to be much more hetero-geneous in their specific molecular effects, e.g. excessive protein stabilisation, toxic aggregation, modulation of protein activity or interaction selectivity[44,46,47,64]. However, we did observe a gen-eral trend of GOF variants occurring in positions characterised by higher degree of disorder, and subsequently looser residue packing, providing one explanation of why the observed stability perturbation is so mild.

Previously, there has been very little consideration of how variant effect predictors perform for pathogenic mutations asso-ciated with different molecular mechanisms. One study found that SIFT and PolyPhen appeared to underperform on GOF mutations, but this involved a very small set of known disease mutations[65]. Our observation that nearly all tested VEPs perform worse on non-LOF mutations was very surprising, and has major implications for the identification of pathogenic variants. Com-putational predictors are very widely used for the prioritisation of potentially causal variants, and if they are systematically under-performing on DN and GOF mutations, this could translate into large numbers of true pathogenic variants being missed by cur-rent sequencing studies. Given our results, there are no specific

predictors we can particularly recommend if DN or GOF mutations are suspected, although we continue to, in general, recommend DeepSequence[53] or other related unsupervised methods based upon our previous benchmarking study[7]. At the moment, we suggest that, if a DN or GOF mechanism is suspected, one should be very careful about filtering out variants due to lower VEP scores.

The clustering of DN and GOF mutations in three-dimensional space presents one possible strategy for improving their prediction. However, this is limited to those genes where multiple disease mutations are already known, which greatly limits the applicability of clustering-based approaches. In addition, even though we see a strong enrichment of clustering in DN and GOF mutations, clustering of pathogenic missense mutations can still be associated with a LOF mechanism. For example, if a protein has multiple folded domains, separated by flexible linkers, and destabilising LOF mutations are most likely to occur in the folded domains, this will result in apparent clustering of LOF mutations. The strategy we have used here may be relatively immune to this, as we only considered structured regions of proteins, but if one uses full-length protein models that include flexible/disordered regions (e.g. AlphaFold models[42]), or a sequence-only clustering approach, then more significant clustering of LOF mutations might be observed.

Due to the size of our dataset and the computational effort required, we explored only one stability prediction method as part of this study. However, other distinct methodologies could potentially reveal extended perturbation effects of stabilising and non-LOF mechanism variants. For instance, DynaMut2[66], a stability prediction method based on normal mode analysis, can evaluate mutation-induced changes to protein dynamics and flexibility, which could be important for understanding the effects of DN and GOF variants in protein complexes. However, the majority of such methods currently offering unique advantages are only accessible as webservers, severely limiting the ability to carry out and troubleshoot large-scale prediction efforts.

Another way of improving the identification of non-LOF mutations is with high-throughput experimental approaches, e.g., deep mutational scanning (DMS). These experiments have shown tremendous power for the direct identification of pathogenic variants[67–71], and if an appropriate experimental phenotype is chosen, it should be possible to measure an appropriate variant effect for DN and GOF mutations. However, not all DMS strategies are appropriate for different types of mutations. A recent study using fluorescent-coupled PTEN to measure the abundance of different variants was able to identify destabilising disease mutations, but not known DN mutations[72–74]. Future work will be required to determine the best ways to systematically identify pathogenic mutations associated with different molecular mechanisms using a DMS-like strategy.

## Methods

**Structural variant dataset collection and annotation.** All mutations, variant and gene annotations, corresponding structure identifiers, predictor and inheritance or mechanisms-based group gene clustering values are accessible through the link found in the 'Data Availability' section.

The pathogenic missense mutations used in this study were downloaded from ClinVar on 2021.01.28, selecting those labelled as 'pathogenic' and 'likely pathogenic'. Putatively benign missense variants were taken from gnomAD v2.1.1. Any variants present in the ClinVar dataset were excluded from the gnomAD set. No filtering for allele frequency was performed in the gnomAD variants, as this would dramatically reduce the size of our dataset. For certain analyses involving recessive mutations, we used only those gnomAD variants that had been observed at least once in a homozygous state.

Protein structures were downloaded from the Protein Data Bank on 2020.08.17, considering the first biological assembly for each structure to maximise the chance that it represents the biologically relevant quaternary structure. Mutations were mapped to protein structures in the same manner as previously described[32], considering those polypeptide chains with >90% sequence identity to a human

protein over a region of at least 50 amino acid residues. While in some cases this includes non-human structures, allowing us to substantially increase the size of our dataset, mutations were only mapped to structures where the residue of interest, as well as its adjacent neighbours, were the same as the human wild-type sequence. In the case where a residue maps to multiple PDB files, we selected a single chain based on sorting by best resolution followed by largest biological assembly. Only the first structure was extracted from NMR ensembles. Importantly, each variant was mapped to only a single residue in a single PDB structure, and residues missing from PDB structures were not considered. For PDB files containing multiple occupancies of a single residue, only the first occurring entry was selected.

Structural locations were classified as interior, surface, or interface according to a previously classification[75]. Interface residues show a solvent-accessible surface area difference between the free subunit and full protein structure. Other residues with less than 25% relative solvent accessible surface area in the full structure were classified as interior, while the remainder was designated as surface.

Molecular disease mechanism annotations for genes were derived based upon information available in the OMIM[29] and ClinGen[33] databases. First, only genes that were annotated with an inheritance of 'AD' in OMIM were considered for dominant mechanism annotation. Those genes annotated as 'Sufficient evidence for dosage pathogenicity' in ClinGen were classified as HI. Next, we searched all OMIM entries for the keywords 'dominant negative', 'gain of function' and 'activating mutation'. Then, we manually read the OMIM entries for all genes identified in this search. If there was evidence in the OMIM entry that a pathogenic missense mutation was due to one of these mechanisms, then it was assigned as DN or GOF. AR disease genes were assigned from OMIM genes associated with disease inheritance marked by the 'AR' category. Importantly, we tried to keep this process as unbiased as possible, so we only considered information available in the OMIM entries. While for many genes, there is further evidence available in the literature regarding molecular disease mechanisms available, this could lead to bias on our part if we spend more time investigating certain genes than others.

Protein functional class annotation was derived based on work published by Iqbal et al.[37]. We expanded their 24-class annotation with the inclusion of a "Translational Protein" class, and produced missing gene labels using PANTHER[38] annotation and manual curation of entries in line with the underlying categories. Multiple functional class labels can be associated with a single gene.

For external validation we derived an independent structural dataset, based on variant-level GOF and LOF disease mechanism annotation by Bayrak et al.[40]. The HGMD GOF and LOF label data was downloaded from https://itanlab.shinyapps.io/goflof/ on 2021.10.25. We used the UniProt[76] identifier mapping API to determine the canonical UniProt accession and primary sequence for each protein. We then mapped the chromosomal position of each variant to the canonical sequence using the EBI protein coordinates API[77]. OMIM 'AD', 'AR' and 'ADAR' genes were used in analyses, unless otherwise indicated. The variants were mapped to AlphaFold models based on UniProt sequence position. AlphaFold models were downloaded from https://alphafold.ebi.ac.uk/download on 2021.07.27. As large AlphaFold structures (over 2700 amino acids; aa) are split up into 1,400aa residue files, protein variants found in multiple files had their FoldX and pLDDT values averaged accordingly. Our ClinVar dataset variants were also reinterpreted in terms of the AlphaFold pLDDT, but with the additional step of deriving pLDDT means at the gene level. This was done to account for and minimise the biases of our gene-level mechanism annotation, and for the variant number disparities between the genes.

HGMD four-class annotation was derived analogously to our ClinVar gene-level classification, with the exception that LOF variants in dominant genes without evidence for haploinsufficiency were labeled as 'Other LOF'. OMIM 'ADAR' genes were excluded to be able to distinguish between dominant and recessive LOF variants, but mixed non-ADAR mechanism genes (HGMD dataset genes containing both GOF and LOF variants in the same gene) were left in the dataset. gnomAD variants from matching genes were included as the putatively benign variant set, excluding variants observed in ClinVar or HGMD disease sets.

**Variant stability and effect prediction.** FoldX 5.0 calculations were performed using all default parameters, essentially as previously described[18]. The procedure was modified to include structures of protein complexes, and the FoldX calculations were set to also take into account intermolecular interactions. Importantly, the 'RepairPDB' function was applied separately to both the monomer and full structures prior to corresponding monomer and full ΔΔG calculations.

Other VEP values were obtained using our previously described pipeline[7]. Where available, VEP predictions were obtained using the dbNSFP database version 4.0[78]. Further predictor scores were download from online sources (PonP2), obtained from predictor web-interfaces (SNAP2, SuSPect, NetDiseaseSNP) or run locally (SIFT, DeepSequence). While for most VEPs we have predictions for the vast majority of mutations considered in this study, due to the high computational requirements of DeepSequence we were only able to run it for 168 proteins. The full list of VEPs employed in this work together with the sample sizes for ClinVar and gnomAD variant predictions can be found in Supplementary Table 2.

**Spatial disease variant clustering.** For the disease mutation clustering analysis, we defined a metric based on the proximity of each protein residue to a known

disease mutation at another residue. For each residue in a protein structure, considering only monomeric subunits, we calculated the Cα:Cα distance $D$ to all other residues with a known ClinVar disease mutation, and the closest distance $D_{min}$ was selected. We calculated the average of the log distance ($\bar{D}$) for all disease residues, and all non-disease residues separately (Eq. 1).

$$\bar{D} = \frac{1}{n}\sum_{i=1}^{n}\log D_{min} \qquad (1)$$

The final clustering metric, which we termed 'Extent of Disease Clustering' (EDC), is presented as the ratio of the two values (Eq. 2):

$$EDC = \frac{\bar{D}_{non-disease}}{\bar{D}_{disease}} \qquad (2)$$

Thus, a value greater than one indicates that the sites of disease mutations tend to be closer to the sites of other disease mutations than non-disease residues tend to be to the sites of disease mutations. One advantage of this simple metric is that it allows clustering to occur in more than one distinct regions of a protein.

In the case of calculating an 'inverse' version of EDC using putatively benign gnomAD mutations, we performed the same procedure by calculating distances to known gnomAD positions and exchanging $\bar{D}_{disease}$ for $\bar{D}_{gnomAD}$ in Eq. 2.

**Statistical testing.** Pairwise statistical comparisons between ΔΔG value groups were carried out using Dunn's test implementation in the R 'ggstatsplot'[79] package, with the p-values for comparisons involving more than two groups being adjusted through Holm's multiple comparison correction.

Dunn's test[80] involves performing $m = k(k-1)/2$ multiple pairwise comparisons using z-test statistics, where $k$ is the total number of groups. The null hypothesis in each pairwise comparison is that the probability of observing a randomly selected value in the first group that is larger than a random value in the second group equals one half, which is analogous to the null hypothesis of the Wilcoxon–Mann–Whitney rank-sum test. Assuming the data is continuous and the distributions are identical except for a difference in centrality, Dunn's test may be understood as a test for median difference[81].

Holm's sequential adjustment[82] is intended to control the maximal familywise error rate and involves adjusting the $m$ p-values of each pairwise test, ordered from smallest to largest. The first p-value is compared to α/$m$, where the alpha level α represents the probability of falsely rejecting the null hypothesis. If the first p-value is found to be less than α/$m$, then it is declared significant and the adjustment procedure continues, with the second p-value now being compared against α/($m$ – 1). Comparisons continue through $i$ ordered p-values until one is found to be greater or equal to α/($m – i + 1$). If a comparison is failed, the procedure stops and all further p-values are rejected[83].

Statistical differences for variant feature proportion ratios were assessed using ggstatsplot or the pairwise Chi-square test implemented in 'rmngb'[84] R package, Cramer's $V$ effect sizes were derived using 'cramerV' from 'rcompanion'[85]. Fraction error bars were derived using the 'binom.confint' function from the 'binom'[86] package. Fisher odds ratios for the stabilising variant analysis were derived using the base R 'fisher.test' function.

ROC analysis was performed in R using the 'pROC'[87] package, with AUC curve differences being statistically assessed through DeLong's algorithm using the 'roc.test' function. Optimal classification thresholds were calculated at default settings using the 'closest.topleft' argument for all proteins with at least 20 gnomAD and pathogenic or likely pathogenic ClinVar variants per-gene, and determined through the closest ROC point distance to the top-left corner of the plot[88]. In the VEP ROC analysis, case-control direction was adjusted individually for each predictor to produce positive predictiveness values above 0.5 AUC.

**Reporting summary**. Further information on research design is available in the Nature Research Reporting Summary linked to this article.

## Data availability

The data generated in this study have been deposited in the OSF database at https://doi.org/10.17605/OSF.IO/H62FQ. Source data are provided with this paper.

Previously published databases or datasets used in this work: gnomAD (https://gnomad.broadinstitute.org/downloads); ClinVar (https://ftp.ncbi.nlm.nih.gov/pub/clinvar/); dbNSFP (http://database.liulab.science/dbNSFP); Itan Lab's GOF/LOF database (https://itanlab.shinyapps.io/goflof/); OMIM (https://www.omim.org/); ClinGen Dosage Sensitivity database (https://search.clinicalgenome.org/kb/gene-dosage).

## Code availability

Code to calculate our mutation clustering metric from PDB files is available at https://doi.org/10.5281/zenodo.6759338.

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

## Acknowledgements
This project was supported by the European Research Council (ERC) under the European Union's Horizon 2020 research and innovation programme (grant agreement No. 101001169), and the Medical Research Council (MR/M02122X/1). J.M. is a Lister Institute Research Fellow. B.L. is supported by the Medical Research Council Precision Medicine Doctoral Training Programme.

## Author contributions
L.G. performed the computational analyses and interpreted the data, under the supervision of J.A.M.; B.J.L. performed the variant effect predictor calculations. L.G. and J.A.M. wrote the manuscript.

## Competing interests
The authors declare no competing interests.
