## [Peer Review File · Nature Communications]

Reviewers' Comments:

Reviewer #1:

Remarks to the Author:

Nature Communications manuscript NCOMMS-21-43599-T

The manuscript from Gerasimavicius et al. presents a data-driven, statistical analysis of "putatively benign" and pathogenic mutations in disease genes associated with different modes of inheritance (autosomal dominant/AD or autosomal recessive/AR) and disease mechanisms (gain of function/GOF, dominant-negative/DN, loss of function/LOF, haploinsufficiency/HI), accounting for the mutations' locations in 3D structures of proteins, and their effect on proteins' stability. As a measure of the predicted effect of mutations on proteins' stability, authors primarily used absolute free energy change upon mutations, calculated using FoldX v5. Additionally, authors evaluated protein-level clustering of mutations to investigate the difference in the clustering of mutations in genes associated with LOF and non-LOF disease mechanism.

The use of proteins structures in characterizing and identifying pathogenic from benign mutations has become increasingly clear over the last couple of years. This study takes it one step ahead by analyzing mutations' effect using protein structures separately for mutations in genes associated with different modes of inheritance and molecular mechanisms. This article is thus timely. I find the result reporting on the utility of considering complete biological assembly of proteins (protein complex structures) in the identification of disease mutations on a large scale, as one of the most profound results of this study.

However, several crucial concerns remain. I find the interpretation of results obtained from gene- or protein-level annotations and analysis as mutation-level results, controversial and misleading, especially when not validated. Please find my comments in detail below:

1. (data/reproducibility) Result (subsection: 1)

Were all 211,266 gnomAD and 13,050 ClinVar mutations consistently mappable to both monomeric and complex structures of 1,261 proteins, as reported in Figure 1? Authors should report the corresponding PDB identifiers for monomeric and complex structures of 1,261 proteins in a supplemental table.

2. (data/reproducibility) Result (subsection: 2)

a) Authors report that to perform the comparative analysis of the effect of recessive and dominant mutations on stability, they used pathogenic ClinVar mutations from 726 autosomal recessive genes and 535 autosomal dominant genes, excluding those with mixed inheritance. But 726 and 535 genes sum up to 1,261 genes, which is the total genes authors have analyzed, according to their reporting in Results, subsection 1, page 3. Then what are the genes that were excluded? I invite authors to clarify.

b) Further, a list of 1,261 genes that have been analyzed along with their classification form OMIM in terms of mode of inheritance: autosomal recessive (726 genes), autosomal dominant (535), mixed (?), and mechanisms: LOF, HI, DN, GOF, must be included as a main or supplemental table. These data will allow for the replication of the results.

3. (result interpretation and reporting) Result (subsection: 2)

a) Authors made a vast generalization in their annotations of mutations as AR and AD, which are basically gene-wise annotations. They have discussed the rationale and necessity behind this generalization in the paper. However, the way they present the results of analyzing these, if not critically flawed then over-generalized at best, data is misleading.

For example, the title of the second subsection of Results: "Recessive mutations are more structurally perturbing than dominant mutations", should be "Mutations in recessive disease genes are more structurally perturbing than those in dominant disease genes." Similar texts throughout the paper, including figure captions, should be coherently corrected in this manner, to avoid any occurrences of misleading.

b) (Figure S2) Authors have analyzed the predicted effect of gnomAD variants on proteins' stability accounting for zygosity. Two questions/comments here:

- Did the authors make sure all 71,022 gnomAD variants from AD genes are heterozygous?

- The comparative effects (as well as non-significant difference) of gnomAD heterozygote variants from AD genes and homozygous variants from AR genes would make sense, as these are putatively benign variants in two different zygosity levels. Besides all variants from AR genes (as the authors reported in Figure S2), I am interested to see how do the gnomAD heterozygous variants from AR genes compare.

c) (Figure 2) What is the zygosity state of 7,371 ClinVar pathogenic mutations in AD genes? All heterozygous? Similarly, what is the zygosity state of 5,679 ClinVar pathogenic mutations in AR genes? All homozygous?

d) (Discussion) Authors have not delved deeper into the data/results to understand and discuss the driving factors behind observing the difference in the ClinVar pathogenic variants' effects on protein stability from AD and AR genes. Two suggestions come to mind:

- Is there any substitution type (polar to hydrophobic or aromatic to aliphatic) that is more (or less) frequent in AD versus AR genes?

- The difference they observe is essentially protein-level difference, not mutation level. Is there any type/class of proteins (e.g. kinases, transporters, DNA-binding proteins) that is more (or less) likely to be encoded by AD versus AR genes?

e) (last paragraph of subsection 2 of Results) Description of the results in Figure 2C is too hurried. Please include some quantifications in the main text to support the "remarkable improvement," i.e., 0.67 vs. 0.71, 0.71 vs. 0.77, which is only apparent after careful investigation of Figure 2C?

4. (result interpretation and reporting) Results (subsection: 3)

a) (data) What are the AD genes associated with HI and DN/GOF? I suggest authors include this annotation in the recommended table in my comment 2b.

b) (Validation required) The difference reported in the effect of mutations in genes associated with DN+GOF mechanism versus those associated with HI+AR (Figure 3A) is compelling. Nevertheless, the claim: "Gain-of-function and dominant-negative mutations have much milder effects on protein structure than loss-of-function mutations", requires mutation-level validation. Multiple studies have been performed on single gene and groups of genes, stratifying LOG and GOF mutations in the same protein (four are listed below). Authors must evaluate the predicted effect of these validated LOF/GOF mutations in the same protein before making this claim:

- <https://pubmed.ncbi.nlm.nih.gov/32801145/>

- <https://www.biorxiv.org/content/10.1101/2021.12.02.470894v1>

- <https://pubmed.ncbi.nlm.nih.gov/34633442/>

- <https://pubmed.ncbi.nlm.nih.gov/34948399/>

c) (Page – 5) "DN mutations are far more likely to be found in complexes ...". This sounds strange. Did authors mean DN mutations are far more likely to be found in proteins (or a certain class of proteins) that form homomeric or heteromeric complexes in their true biological assembly, to perform the function? In that case, please rephrase. Can authors report any quantification on how

likely it is based on their own dataset? What type or class of proteins these are?

5. (result interpretation and reporting) Results (subsection: 4)

a) Authors report that stabilization is more common in HI mutations. But as far as I am concerned, their results essentially reveal that stabilization is more common in certain proteins that are associated with the HI disease mechanism. It is a protein or gene-level observation, taking all mutations from a gene as HI.

Authors explanation for their results is interesting and apparently lends support to my argument. The HI mechanism is more common in a certain class of proteins, i.e. transcription factors, which undergo loss of function upon stabilization, therefore, there is a tendency of observing more stabilization in HI. I recommend that authors must revise the interpretation of their results throughout the paper (also suggested in my comment 3a).

b) In this dataset, how many proteins associated with HI are transcription factors?

c) Connecting to my comments 3d and 4c, authors should check protein-class annotations of the genes studied here, which may explain why stabilization is more or less likely to be predictive of pathogenic variants of proteins associated with LOF (HI/AR) and non-LOF (DN and GOF). It is likely that stabilization is more or less likely to be important for certain classes of proteins, which could have preferences for different types of disease mechanisms (LOF/GOF). In fact, it has been previously shown that protein-class level information adds to the predictive power for the identification of pathogenic versus population variants (<https://www.pnas.org/content/117/45/28201>).

d) (Figure S3C) Authors should report p-values for Fisher's test-based statistical analysis.

e) (Page 6) "In Figure S3A, we plot the distribution for full $\Delta\Delta G$ values different types of pathogenic mutations."

- This sentence reads off.

6. (result interpretation and reporting) Results (subsection: 5)

"Moreover, the LOF mutations also tend to be better predicted than non-LOF mutations by those methods based on nucleotide-level sequence conservation, which utilise no information about amino acid substitution type."

Which VEPs authors are referring to by "nucleotide-level sequence conservation", the last panel of Figure 4? Please clarify. And If so, then authors' interpretation does not hold. I see that by 6 out of 7 VEPs, GOF (non-LOF) is better predicted by HI (LOF). Authors are welcome to correct me if I am missing something.

7. (result interpretation and reporting) Results (subsection: 6)

Multiple concerns:

a) It is not clear from the Result section whether the calculation was based on monomeric structure or complex structure.

b) I suggest evaluating the clustering metric for the same set of proteins but separately using monomeric and complex structures. Would authors expect to see any difference?

c) The values of "n" in Figure 5 indicate the number of proteins, rightly so, and unlike all other figures in the paper. So it should be specifically mentioned in the Figure 5 caption that these are the number of genes/proteins, not mutations, given I am understanding it right.

8. (validation and impact) No independent validation

It is hardly surprising that the effect on proteins' stability is predictive of the pathogenicity of mutations. It, however, is an interesting piece of result that the predictive power of proteins' stability varies for genes associated with different disease inheritance and molecular mechanisms. Thinking of the impact of this result, can authors show the value of reclassifying ClinVar VUSes into LOF/GOF using stability measure ($\Delta\Delta G$) followed by a validation using mutagenesis readouts (maybe using data from MAVEdb database)?

9. (discussion) Authors should discuss some exceptional cases and their impact on their results. For example, a protein/gene that they have considered to having associated with LOF or GOF but have both types of mutations. Similarly, a protein/gene that they have considered to having associated with AD or AR but have a mixed inheritance.

10. (clarity) Introduction, Page 2

"We find clear differences between LOF vs. non-LOF mutations in terms of their structural context, their predicted effects on protein stability, and their clustering in three-dimensional space."

It is not immediately clear what structural context authors are referring to. A wide variety of structure properties (secondary structure, protein-protein interaction, etc.) that define the structural context of a mutation have been analyzed in the literature in an effort to characterize variants. Two studies come to mind: <https://doi.org/10.1073/pnas.1820813116>; <https://doi.org/10.1073/pnas.2002660117>. Did authors refer to the mutations' structural location: surface/interior/interface, by context? If so, then it should be stated in the Introduction for clarity.

11. (writing) Results, subsection 1, Page 3

"While we recognise that the gnomAD dataset will contain some damaging variants, e.g. those that are associated with late-onset disease, population-specific penetrance or are pathogenic under homozygous conditions."

The sentence is incomplete.

12. (technical/clarity) Figures captions

"All pairwise group comparisons are significant (... , Holm-corrected Dunn's test)"

This statement has been used multiple times, specifically in Figures 1, 2, 3, and associated supplementary figures, but was never explained. It was not immediately clear what are the groups and what is this test about. It is possible that I just don't know about it, but for readers like me, I recommend elaborating on this test at least in one figure, Figure 1, for example.

13. (technical) Results, subsection 2, page 4

"The differences in perturbation magnitude across the different mutation groups can be partially explained by their enrichment in different spatial locations ..."

The term "enrichment" can be used when a statistical analysis is performed. The authors' claim is based on the difference in relative percentage of mutations in different spatial locations, and it's a correct claim. But I recommend replacing the term enrichment with "relative frequency" or "relative tendency" or "prevalence".

14. (visualization) Figure 5 and Figure S3

It is hard to distinguish colors used for AR and DN. Please change.

15. (typo) Results, Page - 7

Thaty?

Reviewer #2:

Remarks to the Author:

The authors have presented a nice study on the effect of different classes of disease mutations on protein structure using publicly available resources. Later they compare the predictive performance of existing variant effect predictors on these mutations.

They raise the important issue that most VEPs perform worse on non-loss of function mutations, which has major implications for the identification of pathogenic variants.

They present an evidence based suggestion on how tools could be improved.

Hypothesis building and data analysis is well described and whenever there are limitations in their approach these are openly addressed.

I think this work will be helpful for anyone working in the field of disease mutations and might raise awareness for the need of better prediction methods for dominant negative and gain-of-function mutations.

The manuscript reads very well and is clearly structured. Consecutive steps are motivated very well.

Minor suggestions

1. That said, I found the logic in the very first paragraph ("...However, missense variants that are known to be pathogenic represent only a tiny fraction of those that have been observed in the human population^{3,4}. Thus, identifying missense changes that are likely to be clinically relevant remains a major challenge for diagnosis, and ultimately treatment, of human genetic disease.") difficult to follow and a bit weak. Could be reworded to match the higher standard of the rest of the manuscript.

2. To me, it is not stated strongly enough that mutations from disordered regions are not considered. This should be emphasised more clearly in the beginning of the results section. Additionally it could be helpful to have an initial overview figure about the procedure used (see for example Figure 1A in Wang et al. Three-dimensional reconstruction of protein networks provides insight into human genetic disease. Nat Biotechnol, 2012).

3. It's quite unusual to see a whole subsection of the manuscript ("stabilising mutations may be important for gain of function") devoted to only supplementary figures. Maybe that could be reorganised.

4. Figure 5A: Method is not intuitive to understand. It is not stated clearly enough how the ratio is calculated. In the text it is written that a ratio of 1 means random distribution, while >1 means clustering. Hence it is not clear how to interpret, for instance, values lower than 1 shown in the plot or how strong the clustering is for values bigger than 1.

5. Figure S3A: it is not clear why the actual $\Delta\Delta G$ value is meaningful here, when before it was stated that stability predictors often predict the wrong sign.

6. Figure 3B: The significance of enrichment of DN mutations in interfaces could be evaluated with a Fisher's exact test of monomers vs. protein complexes.

7. Figure S3A: regarding "there are still many stabilising mutations". This is hard to grasp from a box plot. It would be helpful to have the data distributions plotted alongside. This would also help to better follow the rationale that non-LOF mutations are of lower magnitude and hence fall into the wrong category due to the margin of error in prediction.

8. Figure S3C: How big are the groups for 0-1.0 and 1.0-4.0?

9. In context of Figure 2B, it could be interesting to look at cases of genes with mixed inheritance (that had been excluded from the study) (i) as a proof of principle, (i.e. do mutations with different inheritance fall into different regions of the same protein?) and (ii) to have some illustrative examples.

10. Although I understand that this work is focusing on the effect of mutations on protein structure, it would be very interesting to put this into context with mutations in disordered protein regions. This is especially true, because the authors emphasise the importance of non-LOF mutations, which are less likely to disrupt protein folding. Since this might be too complex to be added to the analysis, it could at least find some attention in the discussion. (especially concerning Figure 4 "Thus, it does appear that amino acid residues associated with DN or GOF mutations genuinely tend to show weaker evolutionary conservation than those associated with LOF mutations, thus providing another factor that can explain their poor identification by VEPs." and "DN mutations being more perturbative, and GOF mutations occurring at more conserved positions")

Reviewer #3:

Remarks to the Author:

This work, while providing a comprehensive overview of variant effect predictors, completely overlooks efforts dedicated to understanding different molecular effects of mutations on protein stability, dynamics and affinity to its partners using structural information (other proteins, small molecules, nucleic acids, etc).

Results section: "FoldX, as well as other stability predictors, may be better at predicting the magnitude than the sign of the stability perturbation."

This makes no sense from a predictive method development perspective. It has been shown on multiple occasions that robust methods get the direction of the change right (e.g., being able to distinguish stabilising from destabilising mutations). The aforementioned observation, in my view, just indicates limitations of FoldX as a predictive model.

Authors could have calculated molecular interactions explicitly rather than using a method to predict stability effects as a proxy.

Any filtering based on review quality for ClinVar? (5 stars?)

Could have any potential bias/contamination been introduced in the benign mutation set, given no frequency filter was imposed?

Authors should have used AlphaFold2 structures instead, at least to assess mutation effects on the monomers, and to increase their data set size (rather than compromising in data quality, e.g., with the benign mutation selection). I see very little point in limiting this study to experimental structures given the recent advances made available by AlphaFold2.

No description of how structures were filtered is provided? Missing atoms/residues modelled? Crystallographic artifacts and multiple occupancies removed? Any quality check/filter?

This work solely relies on a single, outdated and very limited method (FoldX) to predict effects of

mutation in terms of Gibbs free energy, which has been outperformed by a whole new generation of methods. Relying solely on FoldX predictions could lead to erroneous conclusions. I would advise the authors to include state-of-the-art methods for the different molecular mechanisms (for stability, mCSM, Dynamut2, DeepDDG, MAESTRO, and for oligomer affinity, there are mCSM-PPI2, MutaBind2 - amongst many other options). This contrasts significantly with the variant effect predictors used, which is quite comprehensive.

SUMMARY OF MAJOR CHANGES:

- We added a schematic representation of the data collection, annotation and variant effect evaluation process (**Figure S1**).
- We expanded upon the annotation published by *Iqbal et al.*¹ and included an exploration of protein functional class annotation in the context of our inheritance and molecular disease mechanism labels, finding that functional class association does not drive the underlying $|\Delta\Delta G|$ variance observed for the distinct molecular disease mechanisms (**Figure S5**). While the separate gene groups demonstrate distinct functional class enrichments (**Figure S4**), we show that protein functional class annotation provides an independent dimension to variant interpretation.
- We included an exploration of several physicochemical property indices for disease variants (**Figure S6**).
- We validated our results using an external dataset of variants from HGMD annotated as gain-of-function (GOF) vs loss-of-function (LOF), based on work recently published by *Bayrak et al. 2021*², which we then mapped to AlphaFold model structures. We compared GOF vs LOF variant groups in the context of the full dataset, only OMIM 'ADAR' genes, and only genes with mixed mechanisms (both GOF and LOF in the same gene; **Figure S7**). We also annotated and analyzed the HGMD dataset analogously to our four-class classification approach (**Figure S10, Figure S11**). All analyses validate our previous results based on gene-level classifications of ClinVar disease variants.
- We explored the ClinVar and HGMD datasets in terms of the AlphaFold pLDDT modelling quality metric, which has been shown to be predictive of structural disorder³. We show that GOF variants are significantly more likely to occur at positions characterized by decreased structural order and packing (**Figure S8**). Nonetheless, such variants at disordered positions constitute the minority of both GOF and LOF disease mutations (**Figure S9**).
- We clarified the methodology for deriving and interpreting our clustering metric. The figure has been updated to improve interpretability at a glance (**Figure 6**).
- We included violin plots for stabilizing mutation $\Delta\Delta G$ (**Figure 4A**).
- We included an exploration of functional protein class into the stabilizing variant analysis (**Figure 4D**).
- We added detailed descriptions of the statistical tests and multiple comparison adjustment procedures to the method section.
- We included results from 4 additional variant effect predictors (VEPs).
- We made numerous text and legend clarifications to address the reviewers' comments

As some figure numbers have changed due to the addition of new figures, in this response we will be referring to them by their current labels.

REVIEWER COMMENTS

Reviewer #1 (Remarks to the Author):

The manuscript from Gerasimavicius et al. presents a data-driven, statistical analysis of "putatively benign" and pathogenic mutations in disease genes associated with different modes of inheritance (autosomal dominant/AD or autosomal recessive/AR) and disease mechanisms (gain of function/GOF, dominant-negative/DN, loss of function/LOF, haploinsufficiency/HI), accounting for the mutations' locations in 3D structures of proteins, and their effect on proteins' stability. As a

measure of the predicted effect of mutations on proteins' stability, authors primarily used absolute free energy change upon mutations, calculated using FoldX v5, Additionally, authors evaluated protein-level clustering of mutations to investigate the difference in the clustering of mutations in genes associated with LOF and non-LOF disease mechanism.

The use of proteins structures in characterizing and identifying pathogenic from benign mutations has become increasingly clear over the last couple of years. This study takes it one step ahead by analyzing mutations' effect using protein structures separately for mutations in genes associated with different modes of inheritance and molecular mechanisms. This article is thus timely. I find the result reporting on the utility of considering complete biological assembly of proteins (protein complex structures) in the identification of disease mutations on a large scale, as one of the most profound results of this study.

However, several crucial concerns remain. I find the interpretation of results obtained from gene- or protein-level annotations and analysis as mutation-level results, controversial and misleading, especially when not validated. Please find my comments in detail below:

1. (data/reproducibility) Result (subsection: 1)

Were all 211,266 gnomAD and 13,050 ClinVar mutations consistently mappable to both monomeric and complex structures of 1,261 proteins, as reported in Figure 1? Authors should report the corresponding PDB identifiers for monomeric and complex structures of 1,261 proteins in a supplemental table.

We have now included a schematic figure more clearly explaining our data collection, annotation and evaluation pipeline (**Figure S1**), and clarified the text at several points.

As described in the first paragraph of the 'Results' section, we started the data collection process by identifying ClinVar and gnomAD genes annotated in the OMIM database as either purely autosomal dominant, or autosomal recessive ('AD' and 'AR', respectively). This was done so that we had a set of genes in which dominant-negative, haploinsufficient and recessive disease variants could be separately classified with higher confidence, which would not have been possible when including OMIM 'ADAR' genes. Thereafter, the variants in the chosen genes were mapped to available PDB structures, which resulted in the structural dataset containing 13,050 ClinVar disease and 211,266 gnomAD putatively benign variants, from 1,261 genes. Note that each variant was only mapped to a single structure: the 'monomer' results represent calculations performed using only the isolated polypeptide subunit, while the 'full' results use the entire biological assembly.

Our full dataset was provided in the 'Data Availability' section. The current updated dataset is accessible at <https://doi.org/10.17605/OSF.IO/H62FQ>. We have now highlighted this information at the start of the results section to make it more accessible.

2. (data/reproducibility) Result (subsection: 2)

a) Authors report that to perform the comparative analysis of the effect of recessive and dominant mutations on stability, they used pathogenic ClinVar mutations from 726 autosomal recessive genes and 535 autosomal dominant genes, excluding those with mixed inheritance. But 726 and 535 genes

sum up to 1,261 genes, which is the total genes authors have analyzed, according to their reporting in Results, subsection 1, page 3. Then what are the genes that were excluded? I invite authors to clarify.

Please refer to our answer to your comment 1. We have now clarified the wording. OMIM 'ADAR' genes were excluded entirely from the data collection procedure, as they would have interfered with our mechanism annotation pipeline.

b) Further, a list of 1,261 genes that have been analyzed along with their classification form OMIM in terms of mode of inheritance: autosomal recessive (726 genes), autosomal dominant (535), mixed (?), and mechanisms: LOF, HI, DN, GOF, must be included as a main or supplemental table. These data will allow for the replication of the results.

We address this in our answer to comment 1.

3. (result interpretation and reporting) Result (subsection: 2)

a) Authors made a vast generalization in their annotations of mutations as AR and AD, which are basically gene-wise annotations. They have discussed the rationale and necessity behind this generalization in the paper. However, the way they present the results of analyzing these, if not critically flawed then over-generalized at best, data is misleading.

For example, the title of the second subsection of Results: "Recessive mutations are more structurally perturbing than dominant mutations", should be "Mutations in recessive disease genes are more structurally perturbing than those in dominant disease genes." Similar texts throughout the paper, including figure captions, should be coherently corrected in this manner, to avoid any occurrences of misleading.

We understand the Reviewer's reservations about our approach to generalizing annotations at the gene level, and have now included an external variant-level GOF and LOF annotation dataset from Bayrak et al. 2021², based on the HGMD database.

Comparing the HGMD dataset with our ClinVar data, we find that only 329 genes overlap (between 1261 ClinVar and 797 HGMD genes). We consider this a suitable independent source to validate our results produced through a generalized gene-level classification approach. Notably, even when annotated at variant-level, most of the genes in the HGMD dataset maintain pure mechanism labels at the gene level – only 76 genes have variants from both GOF and LOF mechanisms, from a total of 797 genes when including 'ADAR' inheritance genes.

Using this dataset, we show that:

- GOF variants are milder than LOF variants, in terms of predicted $|\Delta\Delta G|$ (**Figure S7**). This holds true in multiple contexts: the full dataset; considering only 'ADAR' genes; considering only mixed disease mechanism genes, containing both GOF and LOF variants.
- Excluding 'ADAR' genes, we apply our original annotation strategy of using inheritance, GOF/LOF mechanism labels and ClinGen haploinsufficiency data to annotate the variants into four groups, allowing for cases where a gene has both GOF and LOF variants. The resulting groups are 'HI', 'AR', 'GOF' and 'Other LOF' variants (LOF variants in genes without

additional evidence for haploinsufficiency; the variant-level dataset did not label DN variants separately). We find that for both FoldX and the VEP results, the four-group performance closely resembles what we demonstrated using our ClinVar dataset (**Figure S10, Figure S11**).

We feel this validates our approach, which, compared to the Bayrak et al. publication, also showcases the differences between DN, HI and AR mechanism variants. However, to avoid misleading the reader, we have added a clarification at the start of the results section concerning mechanisms, that for the sake of flow and conciseness of the text we will be generally referring to gene-level mechanism assignments. Specifically, we state that *“For the sake of flow and conciseness, from this point in the text we will be referring to the variants from classified genes directly by the associated mechanism (‘DN mechanism variants’ and not ‘variants from genes associated with DN disease’).”*

b) (Figure S2) Authors have analyzed the predicted effect of gnomAD variants on proteins’ stability accounting for zygosity. Two questions/comments here:

- Did the authors make sure all 71,022 gnomAD variants from AD genes are heterozygous?*
- The comparative effects (as well as non-significant difference) of gnomAD heterozygote variants from AD genes and homozygous variants from AR genes would make sense, as these are putatively benign variants in two different zygosity levels. Besides all variants from AR genes (as the authors reported in Figure S2), I am interested to see how do the gnomAD heterozygous variants from AR genes compare.*

Figure S3 was intended as a technical figure, to show how we account for the large pool of potentially pathogenic recessive variants in a heterozygous state, harboured by the healthy gnomAD subjects. We see that the $\Delta\Delta G$ and structural location prevalence become very similar between the ‘AD’ and ‘AR, Hom’ variants, after accounting for recessive variant zygosity in this dataset. While we cannot control for population penetrance of disease variants mixed in with the putatively benign ones, we can attempt to ensure the fairest comparison of disease and neutral variation in our analyses by controlling this aspect.

In general, variants observed in a homozygous state provide the highest confidence that they are not recessive disease variants or hypomorphic variants in haploinsufficient genes. *We do not see the reasoning behind filtering our homozygous ‘AD’ gnomAD variants. As we focus purely on ‘AD’ and ‘AR’ disease genes, we can have more confidence that the gnomAD variants in ‘AD’ genes are not likely to be pathogenic, at least according to OMIM annotation.*

In regards to ‘AR, All’ vs ‘AR, Het’ variants, there are no significant differences to be observed. As you can imagine from the sample sizes (3,243 homozygous variants being excluded from 140,244 variants in total), the ‘AR, Het’ vs ‘AR, All’ gnomAD variant comparison via Wilcoxon rank-sum test results in p-values of 0.148 and 0.170, for full and monomer $|\Delta\Delta G|$, respectively. The median values for ‘AR, Het’ and ‘AR, All’ are 1.037 and 1.03, while the respective means are 1.887 and 1.873.

c) (Figure 2) What is the zygosity state of 7,371 ClinVar pathogenic mutations in AD genes? All heterozygous? Similarly, what is the zygosity state of 5,679 ClinVar pathogenic mutations in AR genes? All homozygous?

Unfortunately, there is no easy way to obtain the zygosity of the disease variants, as this is not included in the ClinVar and HGMD databases. We can assume that for genes annotated as purely 'AD' in OMIM, that the vast majority of pathogenic variants in ClinVar will be heterozygous. For 'AR' genes, there will be a mixture of homozygous and compound heterozygous variants, but there is currently no way of obtaining this information on a large scale without manual curation.

d) (Discussion) Authors have not delved deeper into the data/results to understand and discuss the driving factors behind observing the difference in the ClinVar pathogenic variants' effects on protein stability from AD and AR genes. Two suggestions come to mind:

Our intention in **Figure 2** was the introductory comparison of 'AD' and 'AR' disease variants as a lead-in to the different underlying mechanisms we have annotated (GOF vs DN vs HI), demonstrating that only some of the dominant mechanisms drive the observed difference. Specifically, we see that the perturbation, structural location and disease variant identification performance differences between HI and AR disease variants are insignificant, as they both in essence are simple loss-of-function mechanisms at the molecular level, even though the former is dominant. In contrast, DN and GOF disease mechanisms appear to be quite complex and demonstrate unique perturbation, location and VEP performance properties.

- Is there any substitution type (polar to hydrophobic or aromatic to aliphatic) that is more (or less) frequent in AD versus AR genes?

We have now included an analysis based on a number of physicochemical property differences observed between the wild-type and mutant variants from different inheritance and molecular mechanism groups (**Figure S6**). While there are some significant differences, no particularly interesting trends are observed, with the exception of the volume change comparison between HI and AR mechanism mutations. While both of the mentioned mechanism variants demonstrate a similar degree of stability perturbation and structural location prevalence, AR mutations cause a significantly smaller increase in residue volume. It could be posited that recessive LOF disease genes are more sensitive to volume changes, leading to the same degree of structural perturbation as the HI variants, which actually induce a higher volume change. We feel delving deeper into this type of analysis, controlling the differences by structural location, could be interesting, but ultimately would distract from the main points of our paper.

- The difference they observe is essentially protein-level difference, not mutation level. Is there any type/class of proteins (e.g. kinases, transporters, DNA-binding proteins) that is more (or less) likely to be encoded by AD versus AR genes?

Thank you for the suggestion to add a dimension of protein functional class to our analysis; it makes sense that particular functions may be associated with specific disease mechanisms. To address this, we have taken and expanded the protein functional class annotation from the Iqbal et al.¹. We indeed see very different functional class prevalence between our mechanism groups (**Figure S4**). Importantly, however, when we explore the predicted stability perturbation effects of variants in the context of associated functional classes vs our disease mechanism groups, we see that these two features are orthogonal. Functional protein class annotation does not explain the stability prediction

heterogeneity we observe between variants associated with distinct molecular mechanisms in majority of the functional classes (**Figure S5**), with the exception of transcription factors.

We do see some interesting deviations from the general trends we observe in **Figure 3A**, with DN variants showing the strongest stability perturbation in a number of functional classes, although this may be influenced by the small sample sizes that arise from partitioning the data on multiple features.

We think both functional class annotation and mechanism annotation demonstrate separate dimensions that could be useful for increasing disease variant identification performance. It would be very interesting in the future to see whether these features, together with others previously explored in literature, could increase prediction performance.

In terms of our paper, we believe putting too much emphasis on the functional class aspect would detract from our intention of shining a light on current method performance issues, seemingly associated with the underlying molecular disease mechanisms. It would be interesting to explore function and disease mechanisms at the variant and domain level, but this would require a more specific data collection setup: separating function at domain or motif level, while having extended multi-class annotation (DN and GOF labels would be of particular interest) for the arising disease mechanisms at variant-level. We are currently unaware of extensive datasets of that kind.

e) (last paragraph of subsection 2 of Results) Description of the results in Figure 2C is too hurried. Please include some quantifications in the main text to support the “remarkable improvement,” i.e., 0.67 vs. 0.71, 0.71 vs. 0.77, which is only apparent after careful investigation of Figure 2C?

We have added the quantitative descriptions of the AUC values into the text body. The error bars in **Figure 2C** are quite narrow and we feel the demonstrated difference is sufficiently striking.

4. (result interpretation and reporting) Results (subsection: 3)

a) (data) What are the AD genes associated with HI and DN/GOF? I suggest authors include this annotation in the recommended table in my comment 2b.

We have addressed this in regard to your comment 1a.

b) (Validation required) The difference reported in the effect of mutations in genes associated with DN+GOF mechanism versus those associated with HI+AR (Figure 3A) is compelling. Nevertheless, the claim: “Gain-of-function and dominant-negative mutations have much milder effects on protein structure than loss-of-function mutations”, requires mutation-level validation. Multiple studies have been performed on single gene and groups of genes, stratifying LOG and GOF mutations in the same protein (four are listed below). Authors must evaluate the predicted effect of these validated LOF/GOF mutations in the same protein before making this claim:

- <https://pubmed.ncbi.nlm.nih.gov/32801145/>

- <https://www.biorxiv.org/content/10.1101/2021.12.02.470894v1>

- <https://pubmed.ncbi.nlm.nih.gov/34633442/>

- <https://pubmed.ncbi.nlm.nih.gov/34948399/>

Thank you for the useful suggestion; we have addressed this in our response to your comment 3a.

c) (Page – 5) “DN mutations are far more likely to be found in complexes ...”. This sounds strange. Did authors mean DN mutations are far more likely to be found in proteins (or a certain class of proteins) that form homomeric or heteromeric complexes in their true biological assembly, to perform the function? In that case, please rephrase. Can authors report any quantification on how likely it is based on their own dataset? What type or class of proteins these are?

We wanted to emphasize that intermolecular interactions are crucial for DN mechanisms, and many theoretical characterizations of the DN effect stem from models in protein complexes.^{4,5} We have now adjusted the wording of this and added a quantitative statistical test: *“Of course, this result is probably influenced by the fact that DN mutations are more likely to be found in protein complexes²², with our dataset showing a significant enrichment of DN-associated mutations in complexes vs monomers ($P = 1.581 \times 10^{-10}$, Fisher’s exact test), compared to the complex-monomer proportion of all other disease variants.”*

If we look at functional class association for genes characterized by DN disorders, we find that the main distinction of complex forming proteins, compared to entries we have only seen as DN-associated monomers, is that they are more often responsible for transporter function (24% vs 6%). This is not surprising; channel protein susceptibility to DN disease mechanisms has been documented in a number of examples^{6,7}. However, our complex vs monomer classification is only based on the available 3D structures, which is a potential source of bias. The overall monomer vs complex functional profile comparison does not yield significant results (Chi-square $p = 0.577$), as we do not have a large number of DN gene examples.

5. (result interpretation and reporting) Results (subsection: 4)

a) Authors report that stabilization is more common in HI mutations. But as far as I am concerned, Their results essentially reveal that stabilization is more common in certain proteins that are associated with the HI disease mechanism. It is a protein or gene-level observation, taking all mutations from a gene as HI.

Authors explanation for their results is interesting and apparently lends support to my argument. The HI mechanism is more common in a certain class of proteins, i.e. transcription factors, which undergo loss of function upon stabilization, therefore, there is a tendency of observing more stabilization in HI. I recommend that authors must revise the interpretation of their results throughout the paper (also suggested in my comment 3a).

b) In this dataset, how many proteins associated with HI are transcription factors?

We were quite curious about the possible HI vs transcription factor association ourselves. Interestingly, using the functional class annotation that we have derived per your suggestion, we actually see that numerically, it is driven by variants from genes in the more general nucleic acid binding protein class (**Figure 4D**). We have included a discussion of these results in the paper.

However, as we have touched upon in our answer to point 3d, we do not see any benefit in reinterpreting all our results through the scope of protein functional class, when there is accuracy to

be gained when combining multiple features to find ways to individualize variant prediction for distinct genes. While some specific protein functions undoubtedly make them more susceptible to specific molecular mechanisms, we have shown in **Figure S5** that these two features represent separate dimensions useful in understanding and improving disease prediction, and do not have to be mutually exclusive. Disease of particular molecular mechanism can also arise independent of a protein's underlying function, with GOF toxic aggregates, or general widespread LOF due to misfolding in essential proteins as just some generic possibilities.

c) Connecting to my comments 3d and 4c, authors should check protein-class annotations of the genes studied here, which may explain why stabilization is more or less likely to be predictive of pathogenic variants of proteins associated with LOF (HI/AR) and non-LOF (DN and GOF). It is likely that stabilization is more or less likely to be important for certain classes of proteins, which could have preferences for different types of disease mechanisms (LOF/GOF). In fact, it has been previously shown that protein-class level information adds to the predictive power for the identification of pathogenic versus population variants (<https://www.pnas.org/content/117/45/28201>).

We have touched upon this in relation to comments 3d, 4c and 5b.

d) (Figure S3C) Authors should report p-values for Fisher's test-based statistical analysis.

We have now included this.

e) (Page 6) "In Figure S3A, we plot the distribution for full $\Delta\Delta G$ values different types of pathogenic mutations."

- This sentence reads off.

Thank you, this has now been corrected.

6. (result interpretation and reporting) Results (subsection: 5)

"Moreover, the LOF mutations also tend to be better predicted than non-LOF mutations by those methods based on nucleotide-level sequence conservation, which utilise no information about amino acid substitution type."

Which VEPs authors are referring to by "nucleotide-level sequence conservation", the last panel of Figure 4? Please clarify. And If so, then authors' interpretation does not hold. I see that by 6 out of 7 VEPs, GOF (non-LOF) is better predicted by HI (LOF). Authors are welcome to correct me if I am missing something.

Thank you for pointing out our error, we have now fixed this and also included a re-interpretation of the result in light of the variant-level dataset performance.

7. (result interpretation and reporting) Results (subsection: 6)

Multiple concerns:

a) It is not clear from the Result section whether the calculation was based on monomeric structure or complex structure.

We have adjusted the text to clarify that monomeric structures were used.

b) I suggest evaluating the clustering metric for the same set of proteins but separately using monomeric and complex structures. Would authors expect to see any difference?

While it would be interesting to explore, especially for homomeric complexes, this approach would pose a number of problems with currently available data. The clustering result would be skewed in heteromeric complexes, where disease variant data may not be available for every subunit. Exploring the cases where disease variant data is available for every subunit of a heteromeric complex would dramatically reduce our statistical power. However, this issue of variant clustering in protein complexes is something we intend to study carefully in the future.

c) The values of “n” in Figure 5 indicate the number of proteins, rightly so, and unlike all other figures in the paper. So it should be specifically mentioned in the Figure 5 caption that these are the number of genes/proteins, not mutations, given I am understanding it right.

We have now included a clarification in the figure legend.

8. (validation and impact) No independent validation

It is hardly surprising that the effect on proteins’ stability is predictive of the pathogenicity of mutations. It, however, is an interesting piece of result that the predictive power of proteins’ stability varies for genes associated with different disease inheritance and molecular mechanisms. Thinking of the impact of this result, can authors show the value of reclassifying ClinVar VUSes into LOF/GOF using stability measure ($\Delta\Delta G$) followed by a validation using mutagenesis readouts (maybe using data from MAVEdb database)?

There are a number of problems with trying to do this. MAVEdb does not contain information as to the mechanism behind reduced fitness scores. Furthermore, while we see a tendency of GOF variants to be milder, there is great overlap in scores derived by our gene-level generalized approach for different mechanisms, as can be seen from the box and violin plots for $\Delta\Delta G$.

9. (discussion) Authors should discuss some exceptional cases and their impact on their results. For example, a protein/gene that they have considered to having associated with LOF or GOF but have both types of mutations. Similarly, a protein/gene that they have considered to having associated with AD or AR but have a mixed inheritance.

We took a generalised approach with a focus on exploring molecular disease mechanisms, and not individual genes. As such, all our hypotheses only involved groups of genes associated with distinct mechanisms. Having hypotheses at the gene level would mean introducing bias to the data collection process itself. For the reasons stated previously, we did not consider mixed inheritance 'ADAR' genes, as early as the data collection stage, as they would have interfered with our generalized mechanism classification process. However, as we have now included a new dataset of LOF vs GOF variants, annotated at the variant and not gene level, this goes some way to addressing the issue.

10. (clarity) Introduction, Page 2

"We find clear differences between LOF vs. non-LOF mutations in terms of their structural context, their predicted effects on protein stability, and their clustering in three-dimensional space."

It is not immediately clear what structural context authors are referring to. A wide variety of structure properties (secondary structure, protein-protein interaction, etc.) that define the structural context of a mutation have been analyzed in the literature in an effort to characterize variants. Two studies come to mind: <https://doi.org/10.1073/pnas.1820813116>; <https://doi.org/10.1073/pnas.2002660117>. Did authors refer to the mutations' structural location: surface/interior/interface, by context? If so, then it should be stated in the Introduction for clarity.

The Reviewer is correct: by structural context we meant structural location. We have adjusted the wording to avoid confusion.

11. (writing) Results, subsection 1, Page 3

"While we recognise that the gnomAD dataset will contain some damaging variants, e.g. those that are associated with late-onset disease, population-specific penetrance or are pathogenic under homozygous conditions."

The sentence is incomplete.

Thank you, we have now corrected this.

12. (technical/clarity) Figures captions

"All pairwise group comparisons are significant (... , Holm-corrected Dunn's test)"

This statement has been used multiple times, specifically in Figures 1, 2, 3, and associated supplementary figures, but was never explained. It was not immediately clear what are the groups and what is this test about. It is possible that I just don't know about it, but for readers like me, I recommend elaborating on this test at least in one figure, Figure 1, for example.

We have clarified this in the legend of **Figure 1**. An explanation and references for Dunn’s test and the Holm multiple-testing adjustment are now included in the methods section, as well as clarified it in the first figure legend with a mention of the methods section.

13. (technical) Results, subsection 2, page 4

“The differences in perturbation magnitude across the different mutation groups can be partially explained by their enrichment in different spatial locations ...”

The term “enrichment” can be used when a statistical analysis is performed. The authors’ claim is based on the difference in relative percentage of mutations in different spatial locations, and it’s a correct claim. But I recommend replacing the term enrichment with “relative frequency” or “relative tendency” or “prevalence”.

We have clarified the figure legends to state that statistical testing was performed in all the figure involving proportion plots. The analyses were performed by Chi-square testing and calculation of Cramer’s V effect size metrics.

We have adjusted the first description of the proportion comparison to specifically mention prevalence, but thereafter we intermix the phrase “enrichment” for flow and variety, as the proportions are indeed different due to enrichments in certain categorical classes.

14. (visualization) Figure 5 and Figure S3

It is hard to distinguish colors used for AR and DN. Please change.

We have adjusted the colours for DN and the new ‘Other LOF’ groups to be more distinguishable.

15. (typo) Results, Page – 7

Thaty?

This has been fixed.

Reviewer #2 (Remarks to the Author):

The authors have presented a nice study on the effect of different classes of disease mutations on protein structure using publicly available resources. Later they compare the predictive performance of existing variant effect predictors on these mutations.

They raise the important issue that most VEPs perform worse on non-loss of function mutations, which has major implications for the identification of pathogenic variants.

They present an evidence based suggestion on how tools could be improved.

Hypothesis building and data analysis is well described and whenever there are limitations in their approach these are openly addressed.

I think this work will be helpful for anyone working in the field of disease mutations and might raise awareness for the need of better prediction methods for dominant negative and gain-of-function

mutations.

The manuscript reads very well and is clearly structured. Consecutive steps are motivated very well.

Minor suggestions

1. That said, I found the logic in the very first paragraph (“...However, missense variants that are known to be pathogenic represent only a tiny fraction of those that have been observed in the human population^{3,4}. Thus, identifying missense changes that are likely to be clinically relevant remains a major challenge for diagnosis, and ultimately treatment, of human genetic disease.”) difficult to follow and a bit weak. Could be reworded to match the higher standard of the rest of the manuscript.

Thank you for highlighting this issue; we have now reworded the paragraph in question to hopefully more clearly state the current issues in variant interpretation, and opening up the reader to the idea that they could be tackled through computational effect prediction methods.

2. To me, it is not stated strongly enough that mutations from disordered regions are not considered. This should be emphasised more clearly in the beginning of the results section. Additionally it could be helpful to have an initial overview figure about the procedure used (see for example Figure 1A in Wang et al. Three-dimensional reconstruction of protein networks provides insight into human genetic disease. Nat Biotechnol, 2012).

We have clarified that disordered proteins or regions are not likely to be included in our ClinVar dataset. However, we have made some adjustments to our datasets and analyses to be able to explore mechanisms in the context of disordered regions, which we expand on in our answer to comment number 10.

In line with your suggestion, we have also included a schematic representation of our data collection, annotation and evaluation process in **Figure S1**.

3. It's quite unusual to see a whole subsection of the manuscript (“stabilising mutations may be important for gain of function”) devoted to only supplementary figures. Maybe that could be reorganised.

This analysis has now been expanded significantly, in particular to include consideration of functional classes, and the figure is now included in the main text (**Figure 4**).

4. Figure 5A: Method is not intuitive to understand. It is not stated clearly enough how the ratio is calculated. In the text it is written that a ratio of 1 means random distribution, while >1 means clustering. Hence it is not clear how to interpret, for instance, values lower than 1 shown in the plot or how strong the clustering is for values bigger than 1.

Thank you for pointing this out. The rare examples of proteins with clustering ratios less than one indicate cases where the sites of disease variants are essentially more evenly distributed throughout the protein than neutral ones, but these are likely to represent chance occurrences due to the nature of the metric rather than meaningful “anti-clustering”. We now explain this in the text. We

have also modified the figure to be more easily interpretable and reworded the method description. Finally, we now make code available to perform the clustering calculations.

5. Figure S3A: it is not clear why the actual $\Delta\Delta G$ value is meaningful here, when before it was stated that stability predictors often predict the wrong sign.

Despite some uncertainty in the prediction of stabilising vs destabilising mutations, we suspect that FoldX still has some overall tendency to get it correct. With panel A we wanted to give an overview of which mechanisms tend to show more distribution towards the stabilising end of the spectrum, including past the -1 – 1 kcal/mol region prone to uncertainty. We have also now replaced the boxplots with violins, to make it more insightful.

6. Figure 3B: The significance of enrichment of DN mutations in interfaces could be evaluated with a Fisher's exact test of monomers vs. protein complexes.

With this panel we wanted to propose an explanation of why the predicted $\Delta\Delta G$ is milder in DN proteins. By definition, the DN effect in complexes requires that the mutant subunits retain the ability to assemble, and thus if majority of mutations occur at interfaces, they would be expected to be mild from a structural perturbation perspective.

We have now clarified that all figures demonstrating proportions represent analyses that were statistically tested with the Chi-square test and Cramer's V calculation. In **Figure 3B** all the Chi-square pairwise comparisons are significant, and in the case of DN vs HI and DN vs GOF, Cramer's V values are 0.26 and 0.2 respectively, representing the strongest observed effects overall.

To validate the observation with a proper enrichment analysis, we carried out a Fisher's exact test for DN variants association with complex membership, as we see a significant enrichment of variants in complexes vs monomers ($p\text{-val} = 1.581e-10$), compared against all other complex-monomer ratios of disease variants not involved in the DN mechanism.

7. Figure S3A: regarding "there are still many stabilising mutations". This is hard to grasp from a box plot. It would be helpful to have the data distributions plotted alongside. This would also help to better follow the rationale that non-LOF mutations are of lower magnitude and hence fall into the wrong category due to the margin of error in prediction.

We have adjusted the panel to include a violin plot, which should make this easier to visualise.

8. Figure S3C: How big are the groups for 0-1.0 and 1.0-4.0?

Bin sample sizes have been added to the figure.

9. In context of Figure 2B, it could be interesting to look at cases of genes with mixed inheritance (that had been excluded from the study) (i) as a proof of principle, (i.e. do mutations with different inheritance fall into different regions of the same protein?) and (ii) to have some illustrative examples.

We have clarified the wording of our data collection pipeline to emphasize we only collected data on 'AD' and 'AR' gene variants. We excluded proteins with mixed OMIM 'ADAR' inheritance at an early stage before data collection, to be able to separate the three molecular mechanism classes of DN, HI

and AR variants. However, with a better approach, it would definitely be interesting to explore both mixed inheritance and mixed mechanism genes, with variant-level assignments including DN mechanism variants. As all our hypotheses were generated at the level of gene groups, we feel specific gene discussion falls outside of the scope of this work, but we could point the reviewer to a review article recently published by our group that extensively discuss molecular disease mechanism examples in different proteins: <https://www.annualreviews.org/doi/abs/10.1146/annurev-genom-111221-103208>

10. Although I understand that this work is focusing on the effect of mutations on protein structure, it would be very interesting to put this into context with mutations in disordered protein regions. This is especially true, because the authors emphasise the importance of non-LOF mutations, which are less likely to disrupt protein folding. Since this might be too complex to be added to the analysis, it could at least find some attention in the discussion. (especially concerning Figure 4 “Thus, it does appear that amino acid residues associated with DN or GOF mutations genuinely tend to show weaker evolutionary conservation than those associated with LOF mutations, thus providing another factor that can explain their poor identification by VEPs.” and “DN mutations being more perturbative, and GOF mutations occurring at more conserved positions”)

Thank you for the suggestion! We have carried out additional data collection to validate our gene-level annotation results, and in the process have included AlphaFold as a structural source. AlphaFold models are annotated with a per-residue modelling quality metric, pLDDT, which notably has been shown to be very predictive of structural disorder³. We have explored our original ClinVar and an external variant-level HGMD GOF vs LOF datasets in terms of the predicted pLDDT vs mechanisms, and found that in both cases GOF variants tend to be associated with significantly lower values, indicating an increased prevalence in less-ordered regions (**Figure S8, S9**).

We have noted this feature as a potential way to identify and GOF variants for individualized interpretation in the discussion section.

Reviewer #3 (Remarks to the Author):

This work, while providing a comprehensive overview of variant effect predictors, completely overlooks efforts dedicated to understanding different molecular effects of mutations on protein stability, dynamics and affinity to its partners using structural information (other proteins, small molecules, nucleic acids, etc).

Results section: “FoldX, as well as other stability predictors, may be better at predicting the magnitude than the sign of the stability perturbation.”

This makes no sense from a predictive method development perspective. It has been shown on multiple occasions that robust methods get the direction of the change right (e.g., being able to distinguish stabilising from destabilising mutations). The aforementioned observation, in my view, just indicates limitations of FoldX as a predictive model.

A primary reason for choosing FoldX was purely practical, as we could not have carried out predictions for a dataset of our size on webserver-based methods. FoldX provides an accessible and parallelizable software solution, that allows troubleshooting by the user, which cannot be said for

webservers. We by no means intended to overlook the methodologically diverse selection of currently available stability prediction methods. In fact, we have previously evaluated 13 different methods, including most of the ones you suggest in your later comment, for the purpose of disease variant identification in the manner of variant effect predictors⁸. However, we did find FoldX to be the most competitive one in terms of performance comparisons with specialized VEPs. As we discuss in that paper, this does not necessarily mean it is predicting changes in stability better than the other methods, but it is supportive of its utility for understanding human disease mutations.

We have now included mention of Dynamut2 in the discussion, as an alternative predictor that we believe could provide a unique perspective on the distinct molecular disease mechanisms.

We did not mean to misattribute the mentioned scale of error to the whole field of stability prediction methods. We have clarified the text to specifically state the error is associated with methods like FoldX and Rosetta, as the source we are citing implies.

Authors could have calculated molecular interactions explicitly rather than using a method to predict stability effects as a proxy.

Our work focuses on exploring the differences between molecular disease mechanisms, some of which involve successful assembly, but manifest disease through poisoning of complexes. We were specifically interested in how the mutation affects the stability of the entire complex, and not just the affinity of molecules at the interface. By involving multiple different tools with separate scoring functions we would not be able to evaluate the performance differences when carrying out predictions on monomer vs complex structures in an unbiased fashion. FoldX uniquely allowed us to perform these analyses consistently in the same framework, with functionality to evaluate protein-protein, protein-ligand and protein-nucleic acid complexes.

Any filtering based on review quality for ClinVar? (5 stars?)

An overwhelming number of ClinVar variants is ranked at only 1 star (criteria provided, either by single submitter or with conflicting interpretation) in the 4-star annotation system, and applying filtering would drastically reduce the dataset and our statistical power.

As an alternative, we have included and reproduced our analyses using an external variant-level annotation dataset based on work published by Bayrak et al.². We used the HGMD variant set, which has separate assertion criteria from ClinVar, and minimal overlap (329 genes overlap, ClinVar and HGMD totals for 'AD' and 'AR' genes are 1,261 and 797, respectively). The analyses (**Figure S7, S10, S11**) validate our previous results.

Could have any potential bias/contamination been introduced in the benign mutation set, given no frequency filter was imposed?

Variant filtering according to the conventional clinical genetics standards (<0.1% of MAF) would drastically reduce the available data, as the vast majority of gnomAD variants would be considered rare. In fact, in light of the findings from the gnomAD data, based on generally healthy individuals, ClinGen have downgraded the PM2 variant interpretation criterion, based on allele frequency in controls, from "Moderate" evidence to "Supporting" (https://clinicalgenome.org/site/assets/files/5182/pm2_-_svi_recommendation_-_approved_sept2020.pdf).

From a practical perspective, gnomAD variation undoubtedly contains recessive or impenetrant disease variants. To achieve a more realistic comparison for recessive disease genes, we compare recessive ClinVar disease variants against homozygous gnomAD variants from matching genes. We find controlling for zygosity in recessive genes brings down gnomAD 'AR' variant features to the same level as gnomAD 'AD' (**Figure S3**).

Authors should have used AlphaFold2 structures instead, at least to assess mutation effects on the monomers, and to increase their data set size (rather than compromising in data quality, e.g., with the benign mutation selection). I see very little point in limiting this study to experimental structures given the recent advances made available by AlphaFold2.

As many of the mechanisms are particularly prevalent in complex proteins, we initially limited our study to targets with available complex structures. We could then more consistently compare predictions performed on monomers to those in the context of the entire complex assembly, thus controlling for the conditions of crystallographic parameters and experimental bias. With this approach we minimize bias and can be more certain variant identification performance was not caused by difference in resolution, crystal packing, etc, but by the differences in structural context between the monomer and the full complex.

However, we have now made use of AlphaFold predicted models for the external HGMD dataset, as well as explored the relationship between pLDDT values and variant mechanisms for both ClinVar and HGMD mutations (**Figure S8, S9**).

No description of how structures were filtered is provided? Missing atoms/residues modelled? Crystallographic artifacts and multiple occupancies removed? Any quality check/filter?

We have now included a more detailed explanation of the pipeline in the Methods section.

This work solely relies on a single, outdated and very limited method (FoldX) to predict effects of mutation in terms of Gibbs free energy, which has been outperformed by a whole new generation of methods. Relying solely on FoldX predictions could lead to erroneous conclusions. I would advise the authors to include state-of-the art methods for the different molecular mechanisms (for stability, mCSM, Dynamut2, DeepDDG, MAESTRO, and oligomer affinity, there are mCSM-PPI2, MutaBind2 - amongst many other options). This contrasts significantly with the variant effect predictors used, which is quite comprehensive.

As we have discussed above, our choice of method was primarily motivated by the scale of our dataset and its performance in our previous benchmarking study. We would absolutely be interested in comparing the prediction results of methodologically distinct tools, like Dynamut2. However, from our previous experience with the mentioned webserver-based methods, evaluating our very large dataset using them would be untenable and troubleshooting of failed predictions would be impossible.

In comparison, the majority of sequence-based variant effect predictors run quite fast, or are already pre-computed on the dbNSFP database for most Uniprot entries; thus we do not agree there is a contrast once the computational effort is taken into account.

FoldX V5.0, which has received an update as recently as 2019⁹, provides functionality which other methods currently lack, enabling us to compare the improvement of variant identification performance using monomer vs complex structures for the various molecular disease mechanism groups. Additionally, is able to evaluate interactions between a wide array of biomolecule types.

On top of interpreting the predicted stability values in terms of mechanism, we also wanted to use a method that showed greatest capacity at actually identifying disease variants based on the produced score, as the most relevant comparison to the various different variant effect predictor methods.

References:

1. Iqbal, S. *et al.* Comprehensive characterization of amino acid positions in protein structures reveals molecular effect of missense variants. *Proc. Natl. Acad. Sci.* **117**, 28201–28211 (2020).
2. Sevim Bayrak, C. *et al.* Identification of discriminative gene-level and protein-level features associated with pathogenic gain-of-function and loss-of-function variants. *Am. J. Hum. Genet.* **108**, 2301–2318 (2021).
3. Tunyasuvunakool, K. *et al.* Highly accurate protein structure prediction for the human proteome. *Nature* **596**, 590–596 (2021).
4. Veitia, R. A. Exploring the Molecular Etiology of Dominant-Negative Mutations. *Plant Cell Online* **19**, 3843–3851 (2007).
5. Bergendahl, L. T. *et al.* The role of protein complexes in human genetic disease. *Protein Sci.* **28**, 1400–1411 (2019).
6. Sottas, V. & Abriel, H. Negative-dominance phenomenon with genetic variants of the cardiac sodium channel Na v 1.5. *Biochim. Biophys. Acta BBA - Mol. Cell Res.* **1863**, 1791–1798 (2016).
7. McEntagart, M. *et al.* A Restricted Repertoire of de Novo Mutations in ITPR1 Cause Gillespie Syndrome with Evidence for Dominant-Negative Effect. *Am. J. Hum. Genet.* **98**, 981–992 (2016).
8. Gerasimavicius, L., Liu, X. & Marsh, J. A. Identification of pathogenic missense mutations using protein stability predictors. *Sci. Rep.* **10**, 1–10 (2020).
9. Delgado, J., Radusky, L. G., Cianferoni, D. & Serrano, L. FoldX 5.0: Working with RNA, small molecules and a new graphical interface. *Bioinformatics* **35**, 4168–4169 (2019).

Reviewers' Comments:

Reviewer #1:

Remarks to the Author:

The manuscript has been significantly revised in the light of the first review, which, has enriched the paper. I have a few additional comments:

Major:

1. Results, subsection 4

The authors performed a commendable job of adding protein functional class annotation into context. It is an important piece of result, shown in Figure S4, which is "there are significant differences in the prevalence of functional classes across inheritance and molecular mechanisms".

Then authors argue that within functional class groups (Figure S5), the general trend of mutation effect on stability holds (i.e., AR > HI > GOF/DN). However, Figure S5 shows that for 4 out of 9 groups, DN variants are most damaging, which makes the title of subsection 4 conflict with what is observed in Figure S5. Authors are welcome to clarify if I am missing something. The authors' explanation for this observation (DN mutations being the most damaging variants for certain protein groups) is reasonable and it makes sense that the protein group annotation is an orthogonal piece of information. Nonetheless, the interpretation of results in the title of subsection 4 and Figure S5 contradict the results per se.

My recommendation would be to revise the title of that subsection to: "There are significant functional class prevalence differences across disease inheritance and molecular mechanisms", which is the clearest result here. And, also revise the title of Figure S5 to: "Underlying functional protein class does not necessarily drive the observed variance differences in distinct molecular mechanism perturbation magnitude".

As far as I am concerned, the analysis of physicochemical properties is not adding any significant value to the manuscript. It is already incredible to see the differential pattern in the stability effect by mutation across different inheritance and molecular mechanisms, which also holds for most protein functional groups/classes. I encourage authors to focus only on this aspect in the 4th subsection of the result for clarity, and conciseness and keep up the focus of the overall manuscript.

(PS: The current sentence stating the header of Figure S5 is incorrect)

2. (Figure 6) One puzzling piece of result is while most of the AR (/LOF) mutations are located in the interior of protein structures, they are rather dispersed in 3D. In contrast, most of the AD (/GOF, DN) mutations are at the surface/interface of protein structures but are clustered. Authors have shown in Figure 3B that gnomAD variants are mostly located at the surface. Do authors expect to see gnomAD variants to be mostly clustered? I suggest the authors add the box plot for gnomAD in both Panel A and B of Figure 6 and discuss how it compares with AD/AR and GOF/DN/HI.

Minor:

My initial concerns about data/reproducibility/validation have been addressed.

I appreciate the authors adding this clarifying statement: "For the sake of flow and conciseness, from this point in the text we will be referring to the variants from classified genes directly by the associated mechanism ('DN mechanism variants' and not 'variants from genes associated with DN disease')."

Additional mutation-specific validation provides endorsement for the results to hold, at least to a

certain extent, in a variant-specific way, not only a gene-specific manner.

The addition of Figure S1 clarified a lot of questions – thanks to the authors.

(technical / clarity) The rest of my comments during the initial review regarding technicality, clarity, and figures have been addressed.

Reviewer #2:

Remarks to the Author:

The authors have addressed all my concerns. In my opinion the manuscript can now be published.

As a side note, I would like to mention that it would have made this process much easier if the authors had pointed out the new numbers of changed figures and where exactly adapted sentences can be found in the text.

SUMMARY OF CHANGES:

- We adjusted naming, figure and text formatting conventions to be more closely in accordance with the Nature Publishing guidelines.
- We adjusted the section and figure titles in relation to **Supplementary Figure 5**.
- We removed **Supplementary Figure 6** and the accompanying text covering physicochemical property changes between inheritance and mechanism groups from section 4, as suggested by Reviewer 1. The remaining figures were renumbered in accordance.
- We named the clustering metric as EDC ('Extent of Disease Clustering') and better described its derivation in the Methods section, with addition of two equations.
- We updated **Figure 6** to also show EDC values if they were derived using not disease but gnomAD variants for the proteins explored in panels **a** and **b**.
- We updated **Figure 6** with an additional panel **c**, which demonstrates concrete structural examples of proteins characterized by EDC values at the opposite ends of the spectrum. We described the panel in a text paragraph, which showcases the utility of EDC for identifying putative non-LOF disease proteins.

REVIEWERS' COMMENTS

Reviewer #1 (Remarks to the Author):

The manuscript has been significantly revised in the light of the first review, which, has enriched the paper. I have a few additional comments:

Major:

1. Results, subsection 4

The authors performed a commendable job of adding protein functional class annotation into context. It is an important piece of result, shown in Figure S4, which is "there are significant differences in the prevalence of functional classes across inheritance and molecular mechanisms".

Then authors argue that within functional class groups (Figure S5), the general trend of mutation effect on stability holds (i.e., AR > HI > GOF/DN). However, Figure S5 shows that for 4 out of 9 groups, DN variants are most damaging, which makes the title of subsection 4 conflict with what is observed in Figure S5. Authors are welcome to clarify if I am missing something. The authors' explanation for this observation (DN mutations being the most damaging variants for certain protein groups) is reasonable and it makes sense that the protein group annotation is an orthogonal piece of information. Nonetheless, the interpretation of results in the title of subsection 4 and Figure S5 contradict the results per se.

My recommendation would be to revise the title of that subsection to: "There are significant functional class prevalence differences across disease inheritance and molecular mechanisms", which is the clearest result here. And, also revise the title of Figure S5 to: "Underlying functional protein class does not necessarily drive the observed variance differences in distinct molecular mechanism perturbation magnitude".

We thank the reviewer for their comments. We have adjusted the titles of the section and the figure according to their suggestion. We indeed mainly wanted to show that functional class grouping serves as an additional orthogonal feature, and not necessarily as the underlying cause for the observed molecular mechanism group differences.

In this and our previous works we have observed high per-protein heterogeneity in terms of the degree of tolerated changes to structural stability. On top of our explanation in the paper, the observed oddly strong perturbations in DN disease proteins could also arise due to intrinsic stability differences between distinct proteins. Outside the cases of full protein unfolding, some protein structural arrangements could be more able to 'buffer' or accommodate perturbations, without causing a full loss-of-function. While others may be more sensitive and unable to maintain function after rearrangement, resulting in lower perturbation values being associated with disease. We speculate this could be especially relevant here as the DN group shows low sample sizes and is more sensitive to outliers when excessively subset by all the different functional class groups.

As far as I am concerned, the analysis of physicochemical properties is not adding any significant value to the manuscript. It is already incredible to see the differential pattern in the stability effect by mutation across different inheritance and molecular mechanisms, which also holds for most protein functional groups/classes. I encourage authors to focus only on this aspect in the 4th subsection of the result for clarity, and conciseness and keep up the focus of the overall manuscript.

(PS: The current sentence stating the header of Figure S5 is incorrect)

We have now removed the figure and the text describing it from the manuscript.

2. (Figure 6) One puzzling piece of result is while most of the AR (/LOF) mutations are located in the interior of protein structures, they are rather dispersed in 3D. In contrast, most of the AD (/GOF, DN) mutations are at the surface/interface of protein structures but are clustered. Authors have shown in Figure 3B that gnomAD variants are mostly located at the surface. Do authors expect to see gnomAD variants to be mostly clustered? I suggest the authors add the box plot for gnomAD in both Panel A and B of Figure 6 and discuss how it compares with AD/AR and GOF/DN/HI.

We have updated the figure to now also include the clustering metric values (which we now call EDC) that were derived using gnomAD variants. While you are correct in noting that gnomAD variants tends to occur most often at the surface, interestingly, they do not cluster as non-LOF disease variants do. They in fact demonstrate the most random-like distribution, with EDC values closest to 1. We do not think this is unexpected, as putatively benign variation should follow a more random pattern. The fact they occur most frequently at the surface should not limit their capacity to not cluster, as protein spatial shapes are irregular (unlike a sphere), especially in the case of multi-domain structures, allowing a balanced distribution in relation to non-mutated positions.

We have now also added a practical example showcasing fringe cases of two proteins associated with EDC values on the opposite ends of the spectrum.

Minor:

My initial concerns about data/reproducibility/validation have been addressed.

I appreciate the authors adding this clarifying statement: “For the sake of flow and conciseness, from this point in the text we will be referring to the variants from classified genes directly by the associated mechanism (‘DN mechanism variants’ and not ‘variants from genes associated with DN disease’).”

Additional mutation-specific validation provides endorsement for the results to hold, at least to a certain extent, in a variant-specific way, not only a gene-specific manner.

The addition of Figure S1 clarified a lot of questions – thanks to the authors.

(technical / clarity) The rest of my comments during the initial review regarding technicality, clarity, and figures have been addressed.

Reviewer #2 (Remarks to the Author):

The authors have addressed all my concerns. In my opinion the manuscript can now be published.

As a side note, I would like to mention that it would have made this process much easier if the authors had pointed out the new numbers of changed figures and where exactly adapted sentences can be found in the text.